# High-density frustrated Lewis pairs based on Lamellar Nb$_2$O$_5$ for photocatalytic non-oxidative methane coupling

Ziyu Chen[1], Yutao Ye[1], Xiaoyi Feng[1], Yan Wang[1], Xiaowei Han[1], Yu Zhu[1], Shiqun Wu[1], Senyao Wang[1], Wenda Yang[1], Lingzhi Wang [1] ✉ & Jinlong Zhang [1] ✉

Photocatalytic methane conversion requires a strong polarization environment composed of abundant activation sites with the robust stretching ability for C-H scissoring. High-density frustrated Lewis pairs consisting of low-valence Lewis acid Nb and Lewis base Nb-OH are fabricated on lamellar Nb$_2$O$_5$ through a thermal-reduction promoted phase-transition process. Benefitting from the planar atomic arrangement of lamellar Nb$_2$O$_5$, the frustrated Lewis pairs sites are highly exposed and accessible to reactants, which results in a superior methane conversion rate of 1456 μmol g$^{-1}$ h$^{-1}$ for photocatalytic non-oxidative methane coupling without the assistance of noble metals. The time-dependent DFT calculation demonstrates the photo-induced electron transfer from LA to LB sites enhances their intensities in a concerted way, promoting the C-H cleavage through the coupling of LA and LB. This work provides in-depth insight into designing and constructing a polarization micro-environment for photo-catalytic C-H activation of methane without the assistance of noble metals.

In recent years, widespread interest has been aroused in converting methane into high value-added chemicals through photocatalytic oxidative or non-oxidative way, which can not only promote the development of clean energy but also effectively achieve the purpose of environmental protection by consuming greenhouse gases[1–10]. Researchers have used various strategies to improve the efficiency of photocatalytic methane conversion, such as introducing other auxiliary oxidants[11–15], heat-assisted photoreaction[16], and improving the separation efficiency of photogenerated carriers[17]. Unfortunately, the photocatalytic methane conversion efficiency is still commonly low, especially for the non-oxidative methane coupling (NOCM) reaction due to the much easier photocatalyst deactivation in the absence of oxidants[18,19], which however is more desirable considering the high carbon-atom efficiency and less CO$_2$ emission. Precious metals have been extensively used for methane activation due to the flexibly tunable geometric and electronic properties[20–25], which however generally face the problems of high cost, easy aggregation, and uncontrollable dehydrogenation. To date, there is still a lack of an efficient strategy for substantially promoting non-oxidative methane conversion over semiconductors without the assistance of noble metals.

A strong polarization environment is the prerequisite for efficient C-H activation[24,26,27]. However, the polarization sites based on adjacent lattice atoms of semiconductor photocatalysts are generally incompetent to activate the C-H bond according to the previously reported low conversion rates[28–30]. Frustrated Lewis pairs (FLP) composed of a sterically encumbered Lewis acid (LA) and Lewis base (LB) pair has flexibly tunable distance and unquenched intensity of LA and LB. Currently, FLP sites have been constructed over defective metal oxides and hydroxides such as CeO$_2$, TiO$_2$, and In$_2$O$_{3-x}$(OH)$_y$ through doping or defect-engineering strategy, which are efficient in promoting water splitting and CO$_2$ reduction[31–34]. It has theoretically proven that FLP can help activate the C-H bond of methane but the efficient practical

[1]Shanghai Engineering Research Center for Multi-Media Environmental Catalysis and Resource Utilization, Key Lab for Advanced Materials and Joint International Research Laboratory of Precision Chemistry and Molecular Engineering, Feringa Nobel Prize Scientist Joint Research Center, Institute of Fine Chemicals, School of Chemistry and Molecular Engineering, East China University of Science & Technology, 130 Meilong Road, Shanghai 200237, China. ✉ e-mail: wlz@ecust.edu.cn; jlzhang@ecust.edu.cn

catalyst has been rarely reported[35,36]. We have recently tried to utilize coupled LA and LB on metal-doped reduced $TiO_2$ for promoting the photocatalytic NOCM, which however fails to break the records achieved by noble metal-containing systems[37]. The inconsistency between the theoretic calculation and practical activity should be attributed to the failure to construct abundant and easily accessible FLP sites during the catalyst preparation. For examples, the FLP formed via doping may be restricted by the doping content; the common undulating surface structure of metal oxides may cause high steric hindrance for producing neighbored but unquenched LA and LB sites. The difficulty of constructing high-density FLP lies in the formation of abundant neighbored but unquenched LA and LB with comparable surface concentrations.

In this work, a thermal-reduction promoted phase-transition strategy is utilized to construct high-density FLP sites on $Nb_2O_5$ composed of low-valence Nb (LA site) and Nb-OH (LB site) that are spatially separated by oxygen vacancy (Vo). The promoted phase transition at low temperatures allows the preservation of Nb-OH while producing abundant Vo, successfully resulting in a high concentration of FLP. A remarkable methane conversion rate of 1456 $\mu mol \, g^{-1} \, h^{-1}$ for NOCM reaction is achieved under light irradiation. The combination of spectroscopic and photoelectronic analyses and theoretic calculation reveals the extraordinary activity attributed to the high accessibility of LA and LB on lamellar $Nb_2O_5$ with photo-enhanced intensity for synergetic C-H activation. This work provides guidance for the rational design and construction of photocatalysts in a highly polarized environment for efficient methane conversion under atmospheric pressure and without additional heating.

## Results

### Phase transitions and defect construction

Herein, lamellar $Nb_2O_5$ with the planar atomic arrangement was selected to construct FLP, which is supposed to reduce the steric hindrance for methane adsorption. Defects are considered to be the key to constructing FLP. Since $Nb_2O_5$ has diverse crystal phases, the structural

distortion and atomic dislocation caused by phase transformation may provide an opportunity to establish abundant defective sites. However, the crystal phase transformation of $Nb_2O_5$ usually relies on high calcination temperature, which may cause severe dehydroxylation, decreasing the possibility to form coupled LA and LB. Therefore, thermal hydrogenation treatment was adopted to promote the phase transformation at lower temperatures considering the efficiency in producing defect sites[38–41], using $NaBH_4$ and $H_2$ as reductants, respectively. The phase structure was investigated by X-ray diffraction (XRD) in Fig. 1a. The pristine $Nb_2O_5$ displays an orthorhombic phase (T-phase). The (0 0 1) and (1 0 1) planes of T-phase $Nb_2O_5$ (JCPDS 30-0873) correspond to main peaks at $2\theta \approx 22.6°$ and $28.3°$ [42]. When annealed with $NaBH_4$, the transformation from T-phase to M-phase (JCPDS No. 72-1484) was initiated at a low temperature of 673 K (673K-$Nb_2O_5$) as characterized by the appearance of a new peak at $2\theta \approx 32.2°$, accompanied by the gradual disappearing peak at $2\theta \approx 28.3°$ [43]. The phase transformation continues with the increasing reduction temperature but the complete phase transformation does not occur even at 973 K (973K-$Nb_2O_5$), although the characteristic peaks of the T-phase become obviously weak. All the $NaBH_4$-treated $Nb_2O_5$ samples show a mixture of dominated T-phase and accompanied M-phase. The positions and intensity of peaks changed along with the phase transformation, which demonstrates the inter-planar change and lattice distortion. In contrast, the phase transformation does not start in the $H_2$ atmosphere even at a high temperature of 873 K (Supplementary Fig. 1), demonstrating $NaBH_4$ is more efficient for inducing phase transformation.

Transmission electron microscopy (TEM) and the element mapping images of the 873K-$Nb_2O_5$ display rod-like structure and uniform distribution of niobium and oxygen elements (Supplementary Fig. 2). The specific surface area (Supplementary Fig. 3) of the $Nb_2O_5$ powder is around 13-15 $m^2g^{-1}$. The lattice distortion is verified by high-resolution TEM (HRTEM, Fig. 1b). The lattice fringes of 0.39 nm and 0.26 nm ascribe to the (0 0 1) planes of T-phase and (2 5 0) planes of M-phase, and obvious lattice distortion fringes are observed[43,44]. As an evaluation of these structural distinctions, the average crystallite size,

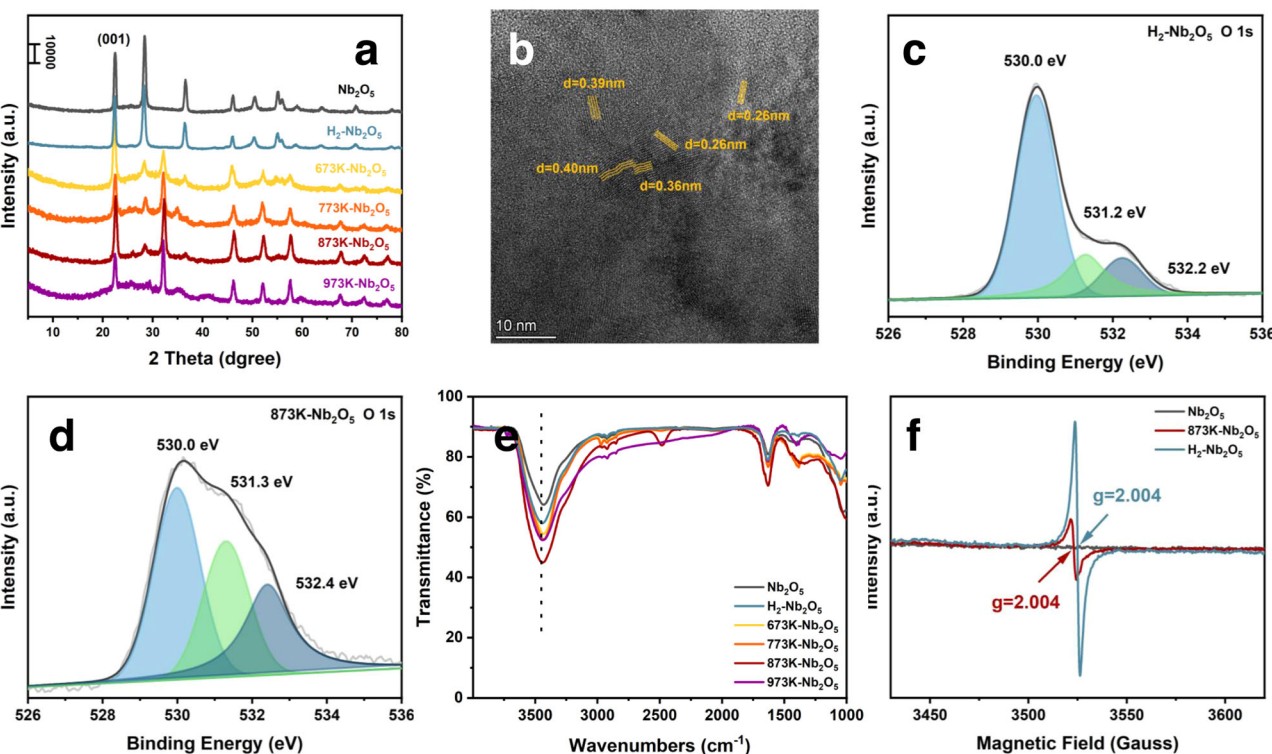

**Fig. 1 | Crystal structure and defect analysis. a** XRD patterns of $Nb_2O_5$ samples. **b** HRTEM image of 873K-$Nb_2O_5$. O 1 s XPS spectra of (**c**) $H_2$-$Nb_2O_5$ and (**d**) 873K-$Nb_2O_5$. **e** FTIR spectra of different $Nb_2O_5$ samples. **f** EPR spectra of $Nb_2O_5$, $H_2$-$Nb_2O_5$, and 873K-$Nb_2O_5$.

dislocation density, and other structural properties were further calculated based on the XRD results[45]. Supplementary Table 1 shows that the dislocation density and strain of $Nb_2O_5$ increase along with the reduction temperature and reach a maximum of 873 K. The dislocation density affords information about the density of structural defects, and strain provides the messages of crystal deformation. Compared with the $H_2$-treated sample, the higher value of $NaBH_4$-treated $Nb_2O_5$ indicates the increased concentration of structural defects due to the formation of mixed phases. The increased lattice strain implies a decrease in the crystal size and stronger lattice distortion. The orthorhombic cell (a = 6.144(2) Å, b = 29.194(3) Å, c = 3.940(4) Å, space group: *Pbam*) was chosen to fit the lattice parameters from XRD results. Supplementary Table 2 shows the pristine $Nb_2O_5$ and $H_2$-treated $Nb_2O_5$ is a relatively standard orthorhombic unit cell; the a-axis and c-axis of the $NaBH_4$-treated $Nb_2O_5$ gradually increase after the various calcination temperature, while the b-axis is significantly shortened. The 873K-$Nb_2O_5$ possesses the maximum deformation of the unit cell, which is consistent with the results of the unit cell strain as calculated above. The above results confirm that during the process of severe lattice stretching and twisting, abundant defects are more possible to generate.

The refined elements X-ray photoelectron spectroscopy (XPS) and electron paramagnetic resonance (EPR) were further used for analyzing the effect of thermal reduction on the electronic state of $Nb_2O_5$. The refined O 1s XPS spectra of 873K-$Nb_2O_5$ exhibit a much higher peak at 531.3 eV corresponding to oxygen vacancy (Vo) sites than that of $H_2$-treated $Nb_2O_5$ at the same temperature (Fig. 1c, d, Supplementary Fig. 4). It is also noted the peak of hydroxyl groups centered at 532.0 eV is significantly improved for 873K-$Nb_2O_5$, which is confirmed by the FTIR analysis as characterized by the broad stretching vibration bands around $3415\,cm^{-1}$ and bending vibration bands around $1648\,cm^{-1}$ (Fig. 1e). The XPS peak of hydroxyl groups decreases with the further increasing of calcination temperature (973K-$Nb_2O_5$, Supplementary Fig. 5). Together with the XRD results, the structure distortion and atomic dislocation of sample 873K-$Nb_2O_5$ from the low-temperature phase transition should result in the simultaneous formation of surface Vo and hydroxyl groups. On the other hand, the EPR analysis indicates that although the signal at g = 2.004 attributed to Vo is improved from the thermal reduction compared with $Nb_2O_5$ (Supplementary Fig. 5, Fig. 1f), sample 873K-$Nb_2O_5$ with the most distorted crystal structure exhibits a lower signal of Vo compared with $H_2$-treated counterpart (Fig. 1c). It is known EPR is only applicable to the detection of single-electron occupied species, so the discrepancy between the EPR and XPS analyses regarding the Vo content should be ascribed to the formation of less single-electron occupied Vo in 873K-$Nb_2O_5$, The most distorted sample of 873K-$Nb_2O_5$ has the highest contents of Vo and hydroxyl groups according to the XPS and FTIR (Supplementary Fig. 4, Fig. 1e), suggesting that the formation of less single-electron occupied Vo in 873K-$Nb_2O_5$ should be related to the concomitant formation of hydroxyl groups. Moreover, the refined Nb 3d XPS spectra indicate 873K-$Nb_2O_5$ has a higher content of low-valence Nb ($Nb_{LV}$) than that of $H_2$-treated $Nb_2O_5$, which should be attributed to the Nb neighboring Vo (Supplementary Fig. 6). It is thus postulated that hydroxyl groups should help promote the electron delocalization from Vo to neighboring Nb, forming more low-valent Nb in sample 873K-$Nb_2O_5$. The above speculation about the interaction with Vo and hydroxyl groups will be corroborated later by combining with other analytic techniques and DFT calculations.

## Photocatalytic performance

The photocatalytic NOCM activity of $Nb_2O_5$ samples was evaluated under light irradiation with a 300 W Xe lamp. Ethane and hydrogen as major products were detected (Supplementary Table 3). A limited amount of propane as the further coupling product was also detected after 4 h, while no oxygenate products were observed. The possibility of thermal catalysis and carbon pollution was precluded through control experiments (Supplementary Fig. 7). Compared with the activity of the $H_2$-treated $Nb_2O_5$, significantly higher activity is found from $NaBH_4$-treated $Nb_2O_5$, which shows a volcano-shaped activity as a function of reduction temperature, with the maximum activity of $1456\,\mu mol\,g^{-1}\,h^{-1}$ on sample 873K-$Nb_2O_5$ (Fig. 2a). This conversion rate is about 22 times higher than that of pristine $Nb_2O_5$ and is more than an order of magnitude higher than all reported non-noble metal photocatalytic NOCM (Supplementary Table 4).

The 873K-$Nb_2O_5$ exhibits stability in the batch reactor when the reaction time is prolonged to 20 h, where the yields of $C_2H_6$ and $H_2$ are continuous and stable (Supplementary Table 5). In stark contrast, methane could not be continuously activated after 8 h over pristine and $H_2$-treated $Nb_2O_5$ (Supplementary Fig. 8 and Supplementary Fig. 9). After 20 h, the conversion rates of $CH_4$ over pristine, $H_2$-treated $Nb_2O_5$ and 873K-$Nb_2O_5$ are 0.1%, 1.2%, and 4.3%. The 873K-$Nb_2O_5$ also presents good cycling stability, with a slight decrease in the yields of ethane and hydrogen, which is possibly due to a small amount of carbon deposition (Supplementary Fig. 10 and Supplementary Table 6).

To further demonstrate the ability of catalysts to activate methane, the NOCM reaction in a flow reactor was carried out (Fig. 2b, Supplementary Fig. 11, and Supplementary Table 7). After a quick induction period, the 873K-$Nb_2O_5$ has a stable ethane production rate of around $22.52\,\mu mol\,h^{-1}$ without a significant decrease within 60 h. The ethane and hydrogen show a stable equimolar ratio during the reaction. The apparent quantum yield (AQY) was further calculated to demonstrate the methane conversion capability of this catalyst (Supplementary Table 8). The AQY of 873K-$Nb_2O_5$ for methane conversion is 0.43% under 365 nm light irradiation and significantly decreases under visible light irradiation.

Elemental analysis (EA) was used to investigate carbon deposits on different samples. Supplementary Table 9 displays the residual coke over different samples. Negligible carbon residual is observed from sample 873K-$Nb_2O_5$, which only increases from 0.6 to 0.76%. The slightly varied balance between hydrogen and ethane should be caused by the gradually accumulated undesorbed product during the cyclic batch reaction. Severe carbon deposition was produced on $H_2$-treated $Nb_2O_5$, which demonstrates that the Vo site alone does not efficiently conduct the process of methane coupling. The TEM, XRD, and XPS analyses of 873K-$Nb_2O5$ after 20 h reaction confirm the sound stability of physicochemical properties (Supplementary Fig. 12).

To understand the enhanced mechanism over $NaBH_4$-treated $Nb_2O_5$, the spectroscopic and photoelectric characteristics of different samples were further analyzed. Compared with the pristine and $H_2$-treated $Nb_2O_5$ only with absorption in the UV region, $NaBH_4$-treated $Nb_2O_5$ shows the photoresponse from UV to near-infrared light regions (Fig. 2c). The photoluminescence (PL) emission, time-resolved fluorescence decay, and transient photocurrent response analyses (Supplementary Figs. 13–15 and Supplementary Table 10) show that the 873K-$Nb_2O_5$ has a good photo-responsive performance. However, there is no obvious difference among different $NaBH_4$-treated $Nb_2O_5$ samples with similar absorption bands, which suggests that the spectroscopic and photoelectric properties are not the main reason for the enhanced NOCM. This result is accordant with the significantly decreased activity of 873K-$Nb_2O_5$ in the visible light region, confirming the enhanced photocatalytic NOCM activity is not only caused by the expanded light absorption.

Infrared spectroscopy of adsorbed pyridine (Py-IR) was further used to understand the contribution of surface functional groups. Figure 2d shows the peak of Lewis acid at around $1450\,cm^{-1}$ is increased through the thermal reduction by $NaBH_4$[46], which should be ascribed to the formation of low-valence Nb adjacent to oxygen vacancy. By combining the results of Nb-OH based on XPS and FTIR, Lewis acid and base sites should co-exist on the surface of $NaBH_4$-treated $Nb_2O_5$. To understand their effect on methane activation, pyridine and pyrrole

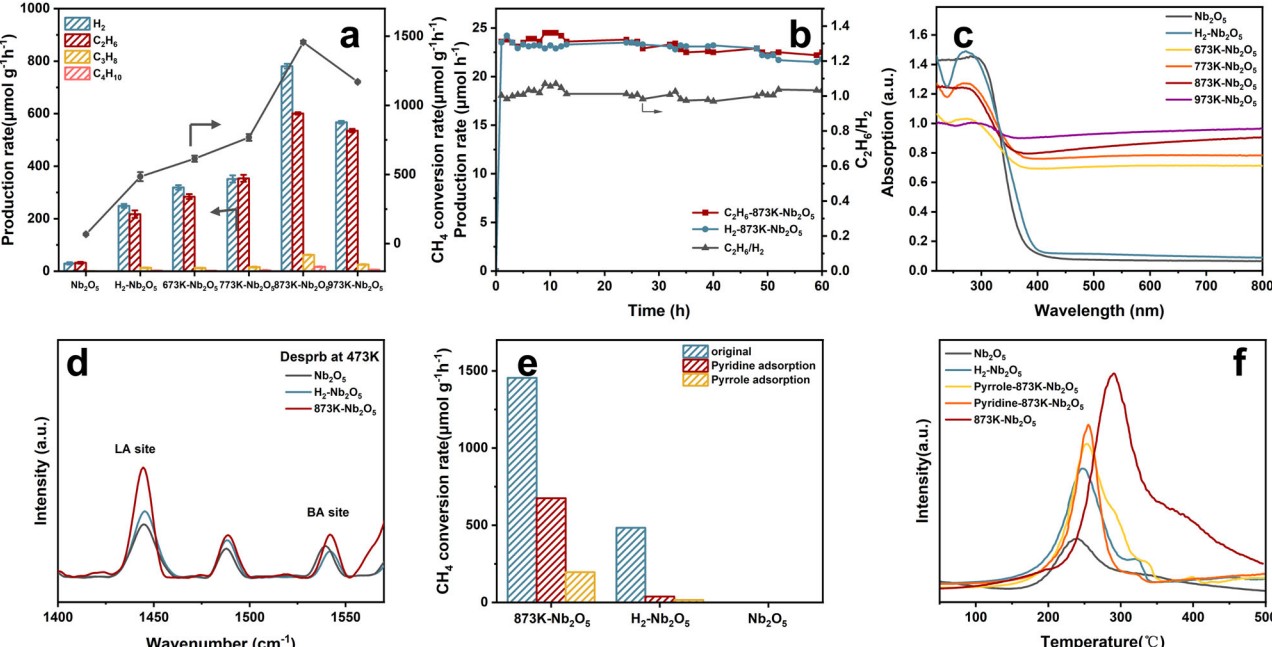

**Fig. 2 | Assessment of NOCM activity. a** The hydrocarbon products, hydrogen yields, and the methane conversion rate over $Nb_2O_5$. Reaction condition: Batch quartz reactor, 5 mg catalyst, 45 mL methane, irradiated under 300 W Xe lamp for 4 h. Shown are mean values and error bars are standard deviation. **b** The long-time photocatalytic NOCM reactions over 873K-$Nb_2O_5$ in the flow-type reactor. Reaction condition: 0.05 g catalyst; photoirradiation area, 28.26 $mm^2$; feed gas, 99.999% of $CH_4$ in flow rate 10 mL $min^{-1}$; SV:55000 $h^{-1}$; irradiated under 300 W Xe lamp for 60 h. **c** UV−vis DRS spectra of different $Nb_2O_5$ samples. **d** Pyridine-IR spectra of different $Nb_2O_5$. **e** The methane conversion rate after pyridine and pyrrole quenching, respectively. **f** $CH_4$-TPD profiles of different samples.

were applied to quench the acidic and basic sites, which lead to significantly decreased activity for different samples (Fig. 2e and Supplementary Table 11), demonstrating the coupling between the LA and LB sites during NOCM reaction. To verify the effect of site quenching, $CH_4$-temperature-programmed desorption (TPD) measurement was carried out to explain the affinity of acid and base site toward methane molecule and the trend of catalytic activities (Fig. 2f and Supplementary Table 12). The broad desorption bands can be observed in the $CH_4$-TPD profiles of different catalysts. The wide gas desorption peaks at the middle temperature range (200–400 °C) correspond to chemical adsorption. Except for 873K-$Nb_2O_5$, most of the methane desorbs in a main peak around 250 °C, and a trace amount desorbs around 330 °C. While the chemical desorption peak area in the higher temperature range increases significantly as compared with 873K-$Nb_2O_5$, indicating that more $CH_4$ is firmly adsorbed on the surface with FLP sites by chemisorption. The additional peak around 400 °C indicates double sites can further enhance the $CH_4$ adsorption capacity through increasing the binding strength for $CH_4$. The desorption yield of methane from 873K-$Nb_2O_5$ is evidently higher than that from sample only with Vo site or sample 873K-$Nb_2O_5$ with quenched acid/base sites. This trend of the amount of adsorbed methane was well matched with the trend of the methane activation after site quench shown in Fig. 2e, suggesting that methane adsorption behavior should be closely related to methane activation. It is thus speculated that the coupling between the LA and LB sites enables $Nb_2O_5$ to continuously activate the C–H bond with refreshed active sites.

## Structural Analysis for verification of FLP sites

X-ray absorption spectroscopy (XAS) was further used to finely explore the local structure of reduced $Nb_2O_5$. The X-ray absorption near edge structure (XANES) features of Nb species formed under different reduction conditions are exhibited in Supplementary Fig. 16. Generally, heavy elements like Nb species with weak internal transitions result in weak pre-edge peaks in the XANES. Besides, the majority of Nb sites with octahedral structures in the T-phase and M-phase would

reduce the s-d transition intensity[47]. The Nb in all samples presents a higher oxidation state than the reference sample of Nb foil, and the average valence of Nb gradually decreases with the increasing reduction temperature as evidenced by the edge shift to lower energy. The extended X-ray absorption fine structure (EXAFS) oscillations of the $Nb_2O_5$ samples exhibit a shoulder peak at 5.6 $Å^{-1}$ after reduction treatment (Supplementary Fig. 17). The gradually increasing shoulder peak indicates the severe disarrangements of the Nb local structure during the phase transformation. The rapid fall to low-frequency oscillations on the reduced samples proves the static disorder of atoms around niobium. The rapid decline of oscillations frequency results in a broad low-r peak in the Fourier-transformed EXAFS (FT-EXAFS) spectra (Fig. 3a). Niobium oxide has a highly asymmetric crystal structure and a wide range of Nb-O bond lengths, which creates beats lowering the amplitude of the total signal in the R-space and increasing the difficulty of the fitting[48]. The low crystalline symmetry of T-phase and M-phase $Nb_2O_5$ creates destructive interference of the backscattered electron waves, leading to non-defined peaks. FT-EXAFS exhibits the average distribution of atoms around Nb atoms in different samples, and the distortion and transformation of crystal phase with rearrangement of atoms result in the average changes reflected in R-space. All samples exhibit the main peak of 1–2 Å corresponding to the Nb-O structure. The direct comparison between $H_2$-treated $Nb_2O_5$ and pristine $Nb_2O_5$ reveals a similar average Nb-O distance. The 873K-$Nb_2O_5$ exhibits the most strongly varied average Nb-O distance compared to pristine $Nb_2O_5$. The complex peaks, gradually appearing in 2.5-4 Å with the increasing reduction temperature, reveals the Nb-Nb interaction. The second nearest neighbor Nb atoms do not line up with original Nb-O structure. This difference indicates that the Nb atoms in $NaBH_4$-treated $Nb_2O_5$ exist in a coordination environment different from that of the $Nb_2O_5$.

Figure 3b exhibits fitting results of the Nb-O coordination numbers (CNs). The $H_2$-treated $Nb_2O_5$ possesses the lowest CN among all the samples; according to its similar Nb-O distance with that of pristine $Nb_2O_5$, the formation of Vo does not cause significant atomic

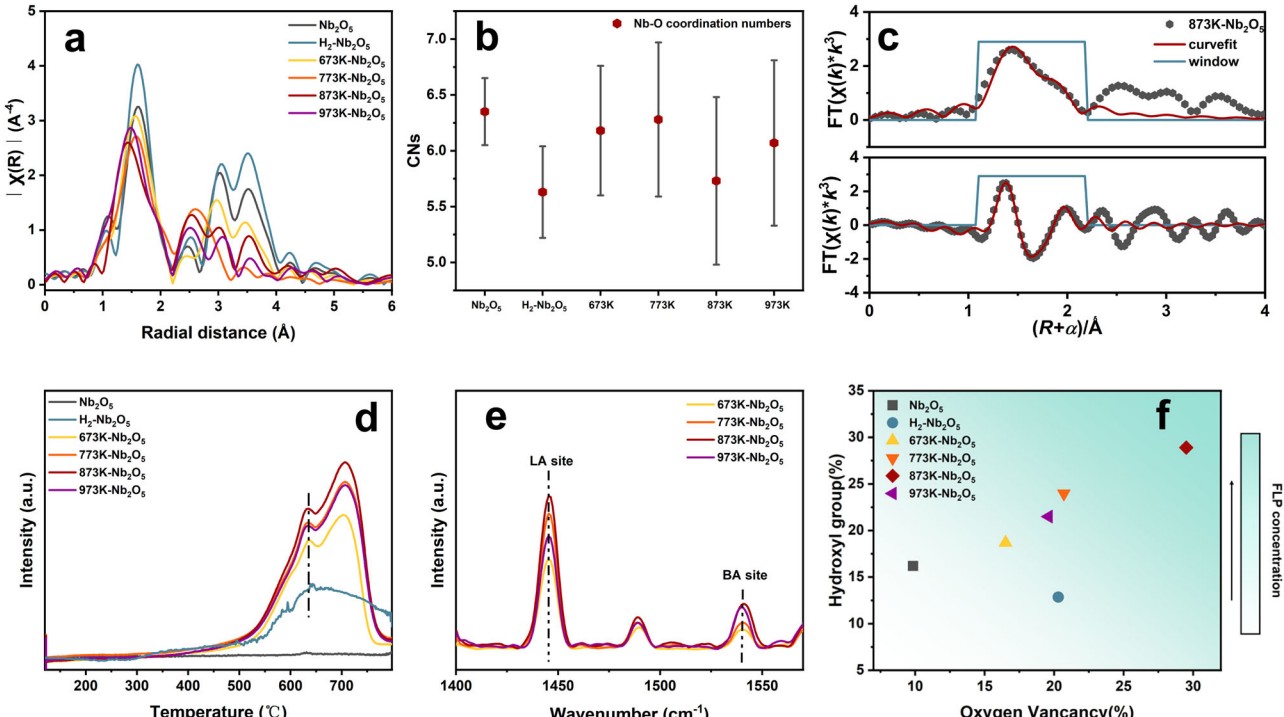

**Fig. 3 | Analysis of FLP sites. a** FT $k^3\chi(R)$Nb K-edge EXAFS of different $Nb_2O_5$ samples. **b** Fourier transformed EXAFS fitting results of the Nb-O coordination numbers (CNs) for different samples. Error bars represent standard deviation. **c** Nb K-edge EXAFS (points) and curvefit (line) for 873K-$Nb_2O_5$, shown in $k^3$-weighted $R$- space (FT magnitude and imaginary components). The data are $k^3$-weighted and not phase-corrected. **d** $NH_3$-TPD spectra of samples. **e** Pyridine-IR spectra of different $NaBH_4$-treated $Nb_2O_5$. (**f**) Relationship between the surface hydroxyl group and oxygen vacancy concentration of different samples.

dislocation. The lowest CN of 873K-$Nb_2O_5$ among $NaBH_4$-treated samples is attributed to the combined effect of plentiful Vo and severe lattice dislocation. Considering asymmetry and distortion in the crystal structure of $NaBH_4$-treated $Nb_2O_5$, and the wide Nb-O distance distribution range, as few paths as possible were chosen to investigate the overall situation of the first coordination shell. The fitting results only represent the average atomic arrangement environment in the sample. Although the fitting results show a simplified average coordination environment, it is supposed that the effect of the reduction treatment on the average Nb-O path can be reflected by fitting. For the cases of pristine and $H_2$-treated $Nb_2O_5$, representative fits can be achieved through one average Nb-O path, while the complex situation of $NaBH_4$-treated $Nb_2O_5$ requires one shorter Nb-O path together with a longer Nb-O path (Fig. 3c, Supplementary Fig. 18 and Supplementary Table 13), confirming the lattice distortion from $NaBH_4$ treatment. To explicitly understand the origin of the longer Nb-O path, the ratio of surface hydroxyl groups is plotted versus the ratio of fitted longer Nb-O path (Supplementary Fig. 19). It is clear that the $NaBH_4$ treatment results in a more significant increase of long Nb-O distance than hydroxyls. Specifically, the percentage of long Nb-O distance increases by about 30% for 673K-$Nb_2O_5$, 773K-$Nb_2O_5$, and 973K-$Nb_2O_5$, while that of the hydroxyl increase for less than 10%, which demonstrates the long Nb-O distance is mainly caused by the structure distortion in these samples. In comparison, 873K-$Nb_2O_5$ shows a further increase of long Nb-O distance and hydroxyl group with a more comparable percentage, which suggests the contribution from hydroxyl to the long-distance Nb-O bond is improved in 873K-$Nb_2O_5$.

Meanwhile, $NH_3$-TPD was further used to reveal the effect of calcination temperature on the acidic characteristics of $NaBH_4$-treated $Nb_2O_5$ (Fig. 3d). The pristine T-$Nb_2O_5$ has negligible acid sites for $NH_3$ adsorption, while all samples treated with $NaBH_4$ show two distinct peaks above 600 °C attributed to strong acid sites. The $H_2$-$Nb_2O_5$ only shows a less intense peak at the lower temperature, suggesting a

weaker acidity. The relation between the reduction temperature and acidity is confirmed by pyridine-FTIR (Fig. 3e, Supplementary Table 14). Through peak fitting of $NH_3$-TPD spectra (Supplementary Table 15), it seems that sample 873K-$Nb_2O_5$ with the most distorted structure has the highest concentration of acid sites. It is also noted the acidity trend of $NaBH_4$-treated $Nb_2O_5$ is accordant with the change of Vo concentration. $NH_3$-TPD was also used to verify the quenching of acid sites through pyridine treatment. The results prove that the strongly acidic sites are indeed covered since the adsorption capacity of the sample for ammonia decreases significantly (Supplementary Fig. 20), which is accordant with the $CH_4$-TPD results. Considering the appearance of $Nb_{LV}$ neighboring Vo from the $NaBH_4$ reduction according to Nb 3d XPS spectra (Supplementary Fig. 6), the $Nb_{LV}$ species with unpaired electron is supposed to function as Lewis acid. Meanwhile, sample 873K-$Nb_2O_5$ shows a comparable increased percentage of Vo and hydroxyls (Fig. 3f), while the other samples have a more obvious increase of Vo. Combining the results from EPR and XPS that the electronic properties of Vo in 873K-$Nb_2O_5$ are altered due to the existence of surface hydroxyl groups, the close correlation between the increased percentages of Nb-OH and Vo in 873K-$Nb_2O_5$ suggests Nb-OH should be a concomitant near Vo. Through the above mutual verification between acidic sites and surface defect groups, it can be known that $Nb_{LV}$ can be used as LA sites, and the adjacent hydroxyl groups can be used as LB sites. It is speculated the adjacent but separated $Nb_{LV}$ and Nb-OH cooperate to promote C-H activation in the way of frustrated Lewis pairs (FLPs), where LA and LB are spatially separated with unquenched acidity and basicity. The function mechanism of unquenched LA and LB pairs will be analyzed and discussed in detail in the theoretical calculation section below.

## Theoretical calculation

The DFT calculation was carried out to verify the existence of FLP composed of $Nb_{LV}$ and Nb-OH and explore its relation with methane

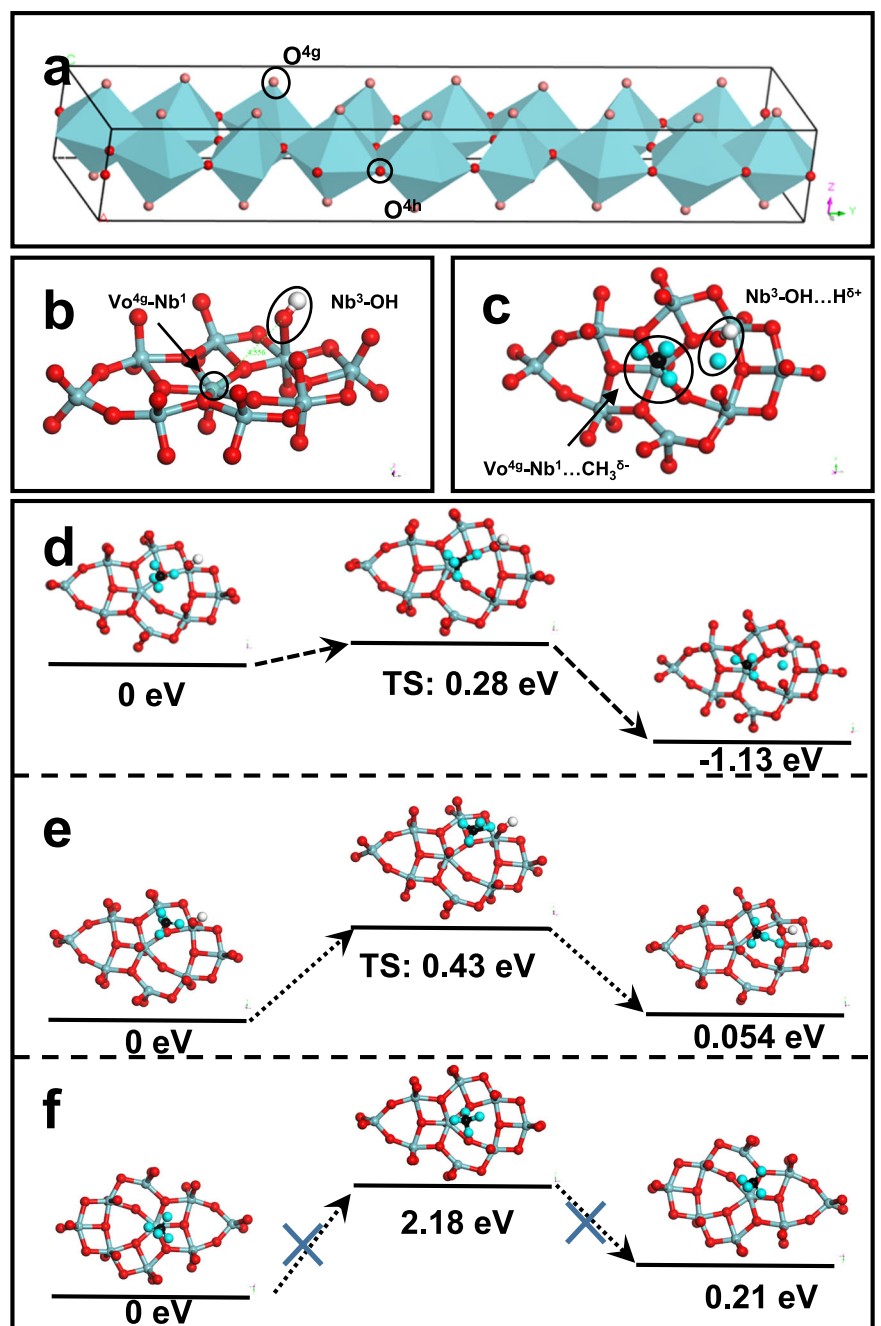

**Fig. 4 | DFT calculations for methane activation on FLP sites. a** Schematic diagram of the unit cell of T-Phase $Nb_2O_5$ containing no partially occupied Nb atoms. The color of $O^{4g}$ is light red, and $O^{4h}$ is red. The Nb atoms are in the centers of different twisted octahedra and pentagonal bipyramids. The color of Nb is blue. **b** Structure of $Vo^{4g}$-$Nb^1$-O-$Nb^3$-OH cluster. **c** Schematic model of methane adsorption on $Vo^{4g}$-$Nb^1$-O-$Nb^3$-OH models. The colors of Nb, O, and C are blue, red, and black, respectively. **b, c** The color of H in the hydroxyl group is white, and in methane is cyan-blue. **d**–**f** Transition states of the lowest energy pathways for C−H bond activation of different models.

activation. T-$Nb_2O_5$ is constructed by highly distorted corner-sharing octahedron and edge-sharing asymmetric pentagonal bipyramid (Fig. 4a). Partially occupied niobium sites (0.8 Nb) in $Nb_{16.8}O_{42}$ unit cell balance the oxygen charge (Partially occupied Nb atom are not shown in Fig. 4a). For the $O^{4h}$-$Nb^{8i}$-$O^{4h}$ structure, $Nb_{LV}$ without a top $O^{4g}$ atom is considered a stronger LA site. Considering the complexity of the T-$Nb_2O_5$ model and the computationally expensive DFT calculation for periodic optimization of a large unit cell, a cluster with a representative structure was selected for theoretical calculation (Supplementary Fig. 21a). The Nb atom in the center of the pentagonal bipyramid structure was named $Nb^1$, and the Nb atoms in the center of the other two octahedral structures in a counterclockwise direction were named

$Nb^2$ and $Nb^3$. The exposed $Vo^{4g}$-$Nb^1$ as LA site forms FLP with adjacent $Nb^3$-OH as Lewis base site (Fig. 4b). The LA site and the LB site are spatially separated by 4.556 Å (Supplementary Table 16). The FLP with sufficient distance provides unquenched LA and LB sites for flexibly stretching the C-H bond of methane. The highly exposed $Nb^1$ site with little steric hindrance also benefits methane adsorption and activation. FLP sites can be formed through six distinct combinations of LA and LB in one incomplete cluster (Fig. 4b and Supplementary Fig. 21b–f), which demonstrates that high-density FLP can be constructed from lamellar $Nb_2O_5$ with the planar atomic arrangement.

It is worth noting that the layers consisting of $Nb^{8i}$ atoms and $O^{4h}$ atoms alternate with the layers of $O^{4g}$ atoms[49]. Contrasting with metal

oxides with the less planar surface atomic arrangement (e.g. (1 0 1) plane of $TiO_2$), the layer composed of $Nb^{8i}$ and $O^{4h}$ in the xy plane makes the FLP sites more accessible to reactants (Supplementary Fig. 22). Each Nb atom has the opportunity to form an FLP site with 6 different hydroxyl groups around it. Theoretically, there is a 53% chance of forming an FLP site on one Nb-O layer per square angstroms. In contrast, there is only a 4% chance for the (1 0 1) plane of $TiO_2$ with the more undulating surface structure to form neighbored Ti and Ti-OH due to the higher steric hindrance. Even if the six-coordinated Ti forms an FLP that satisfies the steric distance, the subsequent methane activation still faces a huge steric hindrance[26]. The distribution of a large number of FLP sites over the $O^{4h}$-$Nb^{8i}$-$O^{4h}$ layer increases the opportunity for methane activation.

To demonstrate the effect of coupling between LA and LB, the models only with LA $Vo^{4g}$-Nb and LB Nb-$O^{4g}$H structures were built for comparison (Supplementary Fig. 23a–f). It is worth noting that the Mulliken charge of Nb atom exposed by single $Vo^{4g}$ is lower than that in FLP models (Supplementary Tables 17 and 18), which proves neighboring Nb-OH group could attract electrons and the electronic properties of Vo was changed. The charge redistribution due to the formation of FLP sites lead to the non-single electron signature at the oxygen vacancies in $NaBH_4$-treated $Nb_2O_5$, and the decrease in signal intensity present in EPR. Although the single LA site is highly accessible, the ability to stretch the C-H bond is comparatively weak. The single LB Nb-OH sites are more efficient than single LA sites for hydrogen abstraction. However, methane cannot be sufficiently polarized due to the lack of charge compensation effects of the single active site. In contrast, all the LA-LB pairs can enhance the ability to activate the C-H bond and abstract hydrogen through charge compensation between LA and LB sites (Supplementary Table 19, Fig. 4c, and Supplementary Fig. 24a–e).

Transition state (TS) calculations for hydrogen abstraction at different sites further approve that FLP sites are thermodynamically favorable for methane activation (Fig. 4d–f). The dissociation of methane over FLP to $CH_3^{\delta-}$ and $H^{\delta+}$ has the lowest barrier of 0.28 eV and the largest exothermic energy of −1.13 eV. It is less thermodynamically and kinetically favorable for a single LA or LB site to activate the methane process. In addition, the TS calculation also proves the synergistic effect of LA and LB sites in FLP. Both transition states of FLP and LB sites feature that the C-H bond is stretched to a sufficient degree. However, the TS from FLP is characterized by a shorter $Nb^3$-OH...$H^{\delta+}$ distance (Supplementary Table 20), demonstrating the higher hydrogen abstraction efficiency. This type of transition state lowers the kinetic barrier for methane activation. Considering abundant FLP sites can be flexibly generated on the (0 0 1) surface of niobium pentoxide, high-density $CH_3^{\delta-}$ groups can be formed, which is supposed to facilitate the subsequent C-C coupling.

### Enhancement of FLP intensity by light excitation

The time-dependent DFT (TD-DFT) calculation was investigated to analyze the effect of light irradiation on the activation of methane at FLP sites in the Vo-$Nb^1$-O-$Nb^3$-OH $Nb_2O_5$ cluster. Considering that the electronic excited state (ES) is formed by the configuration excitation of multiple orbitals, the transition dipole moment density, and the hole-electron analysis were used to understand the charge transfer[50,51]. Supplementary Table 21 lists the bond length and the bond angle of the hydroxyl group of a series of models and Vo-$Nb^1$-O-$Nb^3$-OH in the ground state (GS) and ES, respectively. Compared with the original $Nb_2O_5$ model, the model containing single $Vo^{4g}$ defects or Nb-$O^{4g}$H will cause relatively large interference to the surface structure. Interestingly, for the surface with both $Vo^{4g}$ and surface -OH structures, the two Nb atoms of $Vo^{4g}$-$Nb^1$(LA) and $Nb^3$-OH(LB) that form a Lewis acid-base pair will be closer to each other than the above two models, re-verifying the formation of FLP. For the Vo-$Nb^1$-O-$Nb^3$-OH model, the spatial changes of bond lengths and angles in GS and ES are limited.

The surface structure is not significantly affected by light, which means the influence of photoexcitation on the hydrogen extraction process is based on the tuning of the electron density rather than the distortion of the surface structure. Figure 5a–d exhibits $S_0 \rightarrow S_{75}$, $S_0 \rightarrow S_{83}$, $S_0 \rightarrow S_{139}$, and $S_0 \rightarrow S_{149}$ excitations. According to the electron-hole pair analysis, electrons originally localized near the exposed $Nb^1$ site are transferred to the $Nb^3$-OH group. The excitations could also be studied through the transition dipole moment density, which is shown in the colormap. The main diagonal represents local excitation, and the upper and lower triangles are symmetric matrices, which represent charge transfer. It can vividly reveal the contribution of different atoms to the transition dipole moment, and the electron transfer is obvious. It can be confirmed by TD-DFT that light excitation can both enhance the acidic and basic intensity of FLP, respectively, which is consistent with the results of Py-IR under light irradiation (Fig. 5e and Supplementary Table 22).

The stronger LA and LB sites are supposed to have an enhanced ability to stretch the C-H bond. To verify this, the adsorption energies of $CH_3^{\delta-}$ and $H^{\delta+}$ on $Vo^{4g}$-$Nb^1$ and $Nb^3$-OH were further calculated and compared with the ground state conditions, respectively. Figure 5f and Supplementary Table 23 prove the LA and LB sites become more inclined to adsorb $CH_3^{\delta-}$ and $H^{\delta+}$. The enhanced adsorption of methyl groups is particularly evident at the exposed $Nb^1$ sites, which suggests the bigger charge difference between the $Vo^{4g}$-$Nb^1$ site and $Nb^3$-OH site in dual-active-site catalysts, which could synergistically promote the hydrogen extraction and methyl group adsorption. Therefore, through the photo-induced electron transfer from the LA site to the LB site, the more positively charged $Nb^1$ sites and the more negatively charged $Nb^3$-OH sites upon excitation relative to the ES have greatly enhanced capacity for C-H stretching (Supplementary Fig. 25). The above calculation demonstrates that light irradiation is essential for efficient methane conversion.

In situ infrared absorption spectroscopy was used to verify that methane can strongly interact with FLP. There have been many studies exploring the softened C−H vibrational on the surface of different metal oxides surface[52–55]. Supplementary Fig. 26 shows in situ FTIR spectra after exposing pristine $Nb_2O_5$ and $NaBH_4$-treated 873K-$Nb_2O_5$ to pure methane under dark and light environments. The weakened C−H bond stretching mode typically redshifts from the corresponding gas-phase values (Supplementary Table 24), and the 873K-$Nb_2O_5$ with a more polarized environment under light excitation exhibits the band at 2822 $cm^{-1}$ appeared from the softened stretching vibration $v_1$, which is an infrared forbidden mode in the free $CH_4$ molecules[56]. This peak attributed to soft vibrations confirms more polarized methane activation under photoexcitation. The activation of the $v_1$ mode and the frequency redshift provides the reduction of symmetry of methane over 873K-$Nb_2O_5$[52]. There is no difference before and after adsorption of methane and light exposure over pristine $Nb_2O_5$, which confirms that FLP participates in the polarization of methane and induce the change of C-H bond vibrational mode.

## Discussion

The enhanced NOCM mechanism is proposed based on the above results (Fig. 6). Compared with the pristine and $H_2$-treated $Nb_2O_5$, abundant FLP sites can be constructed on the surface of $NaBH_4$-treated $Nb_2O_5$ through promoted phase transformation at low temperatures. Benefitting from the planar atomic arrangement of lamellar $Nb_2O_5$, the FLP sites in the form of $Nb_{LV}$-Vo-Nb-OH are highly accessible to methane molecules. The unquenched LA $Nb_{LV}$ and LB Nb-OH provide a polarized environment for C-H stretching with flexible distance. The electron transition from LA to LB further enhances the ability of C-H bond activation by light irradiation. The high-density FLP sites allow the efficient C-C coupling of adsorbed -$CH_3$ groups, thereby reducing the occurrence of carbon deposition and improving the utilization of active sites.

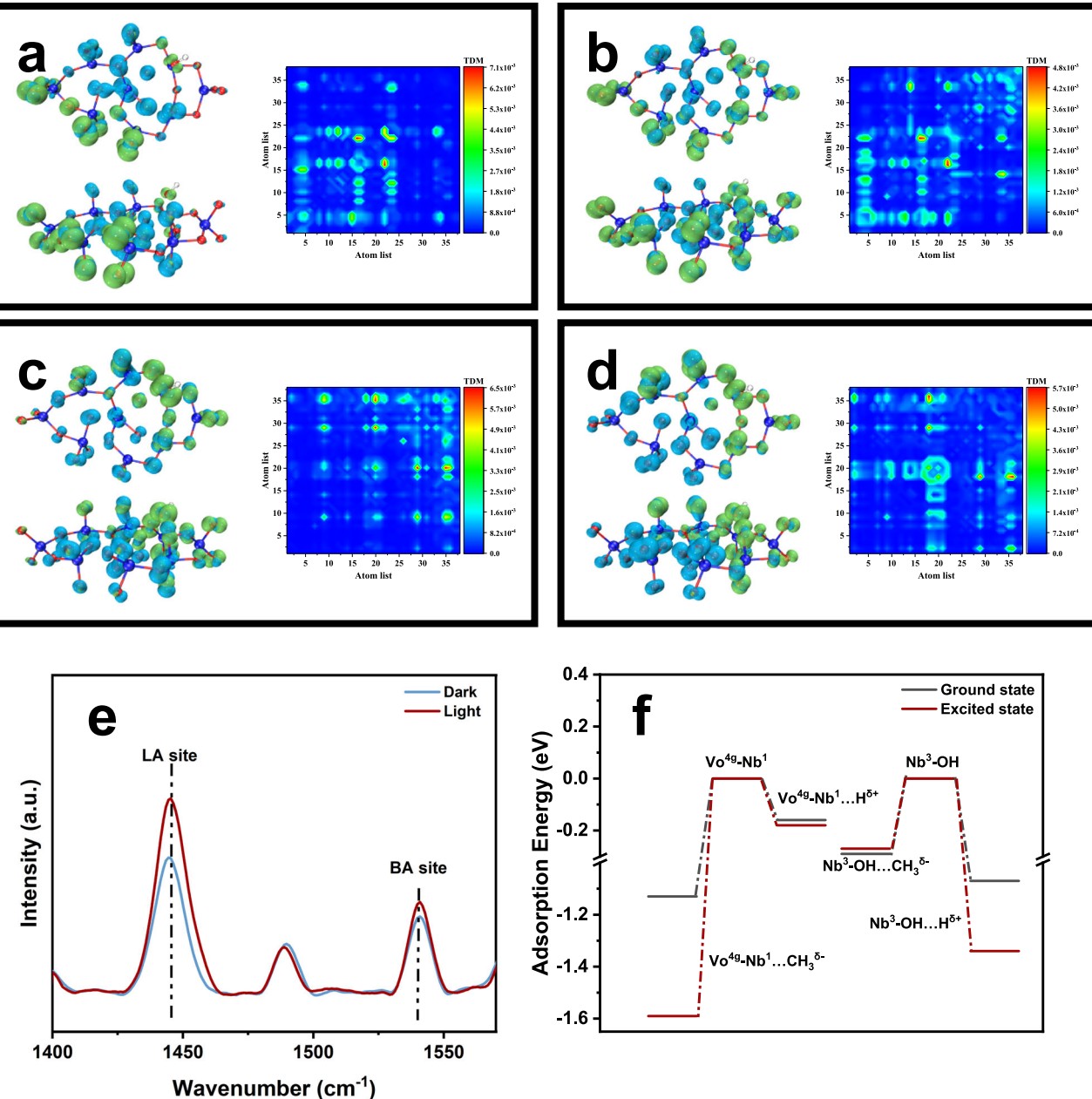

**Fig. 5 | DFT calculations for photocatalytic methane activation.** Analysis of electron-hole pairs of Vo-Nb¹-O-Nb³-OH model in the (**a**)$S_0 \rightarrow S_{75}$, (**b**) $S_0 \rightarrow S_{83}$, (**c**) $S_0 \rightarrow S_{139}$, and (**d**) $S_0 \rightarrow S_{149}$ excitations, respectively. The transition dipole moment density showed in the colormap. **e** Py-IR of 873K-Nb₂O₅ before and after light irradiation. **f** Reaction energy of $CH_3^{\delta-}$ and $H^{\delta+}$ on Vo-Nb¹-O-Nb³-OH model in GS and ES.

The thermal reduction by $NaBH_4$ promotes the phase transformation of $Nb_2O_5$ at low temperatures and generates a large number of defects while retaining abundant surface hydroxyl groups. The unique lattice structure of lamellar $Nb_2O_5$ with planar atomic arrangement allows the abundant hydroxyl groups and $Nb_{LV}$ to form high-density FLP in the way of $Nb_{LV}$-Vo-Nb-OH with high accessibility to reactants. The combination of theoretical calculation and quenching experiments for LA/LB sites reveals the synergy between LA and LB is the key to promoting C-H stretching. The effect of light on strengthening LA and LB intensity through electron transition from LA to LB is revealed through TD-DFT, which results in a high methane conversion rate of 1456 μmol g⁻¹ h⁻¹ for NOCM reaction. Non-oxidative coupling as a model reaction is beneficial to explicitly understanding the relationship between methane activation and light irradiation. Constructing a polarized environment should also be conducive to the improvement

of the activity in other methane conversion reactions, such as methane dry reforming and methane steam reforming. This research demonstrates the principle of constructing a polarization environment for photocatalytic C-H activation of methane, providing a new perspective on the structural design of efficient photocatalysts for methane conversion without the assistance of precious metals.

## Methods
### Preparation of catalysts
The pristine $T$-$Nb_2O_5$ material was synthesized by recrystallization of commercial niobium pentoxide. First, 4 g $Nb_2O_5$ was dispersed in 25 mL HF in a 50 mL Teflon-lined stainless autoclave, the dissolved niobic acid solution was obtained by hydrothermal reaction at 393 K for 4 hours. The white precipitate was obtained by adjusting the pH of the niobic acid solution to 9.0 with ammonia solution, then the operation include

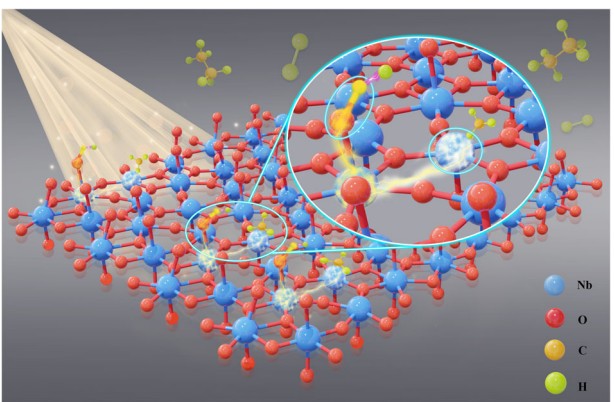

**Fig. 6 | Schematic diagram of Photocatalytic NOCM mechanism.** Photoexcitation causes electron transfer from LA to LB site, which enhances the intensity of LA and LB sites, facilitating the polarization of methane and the activation of C–H bond. The colors of Nb, O, C, and H atoms are blue, red, yellow, and green, respectively.

washing with ultrapure water, and centrifugation was repeated until the supernatant was neutral. The obtained white blocks were calcined at 773 K for 4 h with a heating rate of 2 K/min. The obtained samples were collected and washed with water three times and denoted as $T$-$Nb_2O_5$. The $H_2$-$Nb_2O_5$ was prepared by reduction at 873 K for 4 hours under a 10% hydrogen atmosphere. The FLP-$Nb_2O_5$ was prepared by sodium borohydride thermal reduction treatment. 0.6 g $T$-$Nb_2O_5$ and 0.15 g $NaBH_4$ were mixed and ground well in a mortar, then the mixed samples were calcined at different temperatures for 4 h with a heating rate of 2 K/min. The obtained dark solid was washed with 1 M HCl solution to remove the residual $NaBH_4$ and water to remove HCl.

### Nonoxidative coupling of methane reaction test
**Batch quartz reactor reaction conditions.** First, the catalysts were evacuated in a tube furnace at 393 K in a vacuum environment to remove the adsorbed water and other molecules. 5 mg catalysts were laminated to a closed quartz reactor (45 $cm^3$, photoirradiation area, 28.27 $cm^2$), then the reactor was evacuated for 10 min to remove air. 45 mL of pure methane (99.99%) was injected into the reactor by a gas injection needle and the reactor was placed in dark condition for 1 h to achieve an adsorption-desorption balance. The reactor was irradiated by a 300 W Xe lamp with 2000 $mW/cm^2$ optical power density for 4 h. The methane conversion proceeded under atmospheric pressure and without additional heating (The light band of the lamp, 200–2500 $cm^{-1}$; reaction temperature, 67 °C; the photoirradiation area, 28.27 $cm^2$). The hydrocarbon products were extracted by the gas injection needle and then analyzed by gas chromatography (GC) with a flame-ionization detector (FID). Hydrogen was analyzed by GC with a high-sensitivity thermal conductivity detector (TCD). For a long time reaction, the production was collected every 4 h and replenished to atmospheric pressure with Ar gas after sampling. For the cycle reaction, the sample repeated the vacuum activated after the reaction to ensure that the adsorbed gas molecules were removed before proceeding to the next test.

**Mobile-type reactor reaction conditions.** The catalyst powder was pressed under 40 MPa pressure and ground into 40–60 mesh. Photocatalyst in a quartz cell, 0.05 g; photoirradiation area, 28.26 $mm^2$; cell volume, 113.04 $mm^3$; feed gas, 99.999% of $CH_4$ in flow rate 10 mL $min^{-1}$; SV:55000 $h^{-1}$; light intensity, 2000 $mW$ $cm^{-2}$, reaction temperature, 91 °C.

### Characterization
The morphology was characterized by transmission electron microscopy (TEM, JEM1400) and high-resolution transmission electron

microscopy (HRTEM, JEM2100) at 200 kV. The elements mapping was characterized by scanning transmission electron microscopy (STEM, JEOL 2100 F). Powder X-ray diffraction (XRD) characterization was performed on a Rigaku D/MAX 2550 diffractometer (CuK radiation, $\lambda = 1.5406$ Å) operating at 40 kV and 40 mA and collected data in the range of 5–80° (2θ). The BET surface area measurement was performed by $N_2$ adsorption at 77 K using an ASAP2020 instrument. The UV–vis absorbance spectra of the dry-pressed disk samples were acquired using a Scan UV–vis spectrophotometer (Varian, Cary 500) between 200 and 800 nm, using $BaSO_4$ as the reflectance sample. Photoluminescence (PL) emission spectra of the solid catalysts were obtained using luminescence spectrometry (Cary Eclipse) at room temperature at an excitation wavelength of 350 nm. Transient photocurrent response tests were carried out on a ZAHNER PP211 electrochemical station with a three-electrode cell. The working electrode is made of FTO glass and the sample, the counter electrode was Pt and the reference electrode was a saturated calomel electrode. The $Na_2SO_4$ aqueous solution (0.5 M) was added as the electrolyte, using 470 nm as the excitation light source at temperature. X-ray photoelectron spectroscopy (XPS) was performed on a Perkin Elmer PHI 5000 C ESCA system with Al Kα radiation operated at 250 W. The shift in the binding energy owing to the relative surface charging was corrected using the C 1 s level at 284.6 eV as an internal standard. Electron Paramagnetic Resonance (EPR) was performed on a JEOL-FA200 instrument (fq100.00 md0.35 × 1 am5.00 × 100 tc0.03, test at room temperature, g factor is used for comparison due to the weak change of magnetic field). The elemental analysis was characterized by Elementar Vario EL. The CHNS/CHN mode is used for the test, and the sample is completely burned in an oxidation tube with a pure oxygen atmosphere at 1423 K to generate $CO_2$, $H_2O$, $NO_x$, $SO_2$, $SO_3$, and other gases. Subsequently, the mixed gas was further reduced to $CO_2$, $H_2O$, $N_2$, $SO_2$, and other gases in a reduction tube (1123 K, reduced copper), separated by an adsorption-desorption column, and then separated by a chromatographic column for thermal conductivity detection to obtain contents of elements C, H, N and S.

### Detection of $CH_4$ adsorption
The $CH_4$ temperature-programmed desorption ($CH_4$-TPD) measurements were carried out in a conventional flow system. 50 mg of each catalyst was loaded into the U-shaped quartz reactor. It was reduced at 200 °C for 1 h with a stream of He (30 ml/min). The reactor was then cooled to room temperature and 20 ml of methane was then pulsed into the reactor every minute under a flow of helium (5 ml/min) until the surface of the catalyst was saturated with methane. After purging the reactor with He flows (30 ml/min) at 50 °C for 1 h to remove physisorbed methane, the temperature of the reactor was increased from room temperature to 800 °C at a heating rate of 10 °C/min under a flow of helium (10 ml/min). Desorbed species were detected using a fully automatic chemical adsorption instrument(BelCata II, Japan).

In situ diffuse reflectance infrared Fourier transforms spectroscopy (DRIFTS) was carried out on a Bruker infrared spectrometer (Tensor II) equipped with a liquid nitrogen-cooled mercury-cadmium telluride (MCT) detector in a three-window diffuse reflectance cell (Harrick) with a non-modified dome cover. Two IR measurement windows were made of ZnSe, while the silicon oxide window was used for light irradiation. The resolution was 4 $cm^{-1}$. The IR scanning range was 4000–600 $cm^{-1}$ averaging over 100 scans.

### Acid-base site detection
**Ammonia-temperature-programmed desorption experiment.** The $NH_3$-TPD curve was measured on a fully automatic multifunctional adsorption device. 0.1 g of the catalyst was placed in a device filled with argon atmosphere, first heated to 773 K at a heating rate of 10 K/min, and then pretreated for 1 h. It was then cooled to 393 K, and ammonia gas was adsorbed for 30 min. Under the condition of the helium flow

rate of 50 mL/min, the temperature of the reaction cell was heated from 393 to 973 K at a heating rate of 10 K/min, thereby obtaining the ammonia-temperature programmed desorption curve.

Pyridine monitored by Fourier Transform infrared spectroscopy (FTIR) was performed using an FTIR-650 spectrometer. In a home-made vacuum infrared cell with $CaF_2$ windows, a self-supporting wager of the sample (about 10 mg) was initially dried under vacuum at 623 K for 1 h, and then cooled down to 323 K. Afterward, the wafer was saturated with about 25 mbar of pyridine vapor at 323 K for 10 min and then evacuated again for 30 min to fully remove physisorbed pyridine. Finally, the evacuated sample containing chemisorbed pyridine was subjected to TPD for 30 min, and the IR spectra were recorded in situ situation.

**XAFS measurements conditions.** The X-ray absorption fine structure spectra (Nb K-edge) were collected at the BL14W beamline in Shanghai Synchrotron Radiation Facility (SSRF). The storage rings of SSRF were operated at 3.5 GeV with a stable current of 200 mA. Using Si(111) double-crystal monochromator, the data collection was carried out in fluorescence mode using a Lytle detector. All spectra were collected in ambient conditions.The data reduction and analysis of the XAFS spectra were conducted using the Demeter software package (ATHENA and ARTEMIS, respectively)[57]. All fits were performed in the R space with a k-weight of 3. We applied the Continuous Cauchy Wavelet Analysis (CCWT) to the EXAFS spectra which allows determining the identity of the atoms in noisy signals[58,59].

**Theoretical calculations**

Calculations for the optimized geometries energy and electronic structure were performed using the Dmol3 program within the framework of DFT[60,61]. The ultrasoft pseudopotential was used for electron-ion interactions, and the Perdew-Burke-Ernzerhof (PBE) form of the generalized gradient approximation (GGA) was employed to describe the exchange-correlation functional[62]. All quantum chemical calculations are performed with Gaussian 16[63]. The molecular structure is optimized using the B3LYP functional[64] in the DFT method[65] in combination with the def2TZVP basis set, SDD pseudopotentials for Nb and DFT-D3 correction method[66]. The Excited states calculations with the same level and all Configuration coefficients. The electron-hole pair analysis is implemented by the Multiwfn-3.8 program and is drawn using VMD-1.9.3[51,67–69].

## Data availability

All data supporting the findings of this study are available in the article and its Supplementary Information.

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

## Acknowledgements

This work was supported by the National Key R&D Program of China (2022YFE0107900, 2022YFB3803600, 2021YFC2103501), National Natural Science Foundation of China (22006038, 21972040), Shanghai Municipal Science and Technology Major Project (2018SHZDZX03), the Program of Introducing Talents of Discipline to Universities (B20031, B16017), Innovation Program of Shanghai Municipal Education Commission (2021-01-07-00-02-E00106), the Science and Technology Commission of Shanghai Municipality (21ZR1417900, 20DZ2250400, 22230780200), Shanghai Sailing Program (22YF1410200) and Fundamental Research Funds for the Central Universities (222201717003).

## Author contributions

J.Z. and L.W. conceived and designed the project. Z.C. performed the experiments and analyzed the data. L. W. and J. Z. gave suggestions on the research. Z.C. and L. W. co-performed the DFT calculations and wrote the manuscript. Y.Y., X.F., Y.W, Y.Z, X.H., Sh.W., Se.W. and W.Y. conducted part of photocatalytic measurements. J.Z. and L.W. supervised the project. All authors commented on the manuscript.

## Competing interests

The authors declare no competing interests.
