## [Peer Review File · Nature Communications]

High-density Frustrated Lewis Pairs Based on Lamellar Nb₂O₅ for Photocatalytic Non-Oxidative Methane CouplingREVIEWER COMMENTS

Reviewer #1 (Remarks to the Author):

This manuscript shows a discovery of a new Nb₂O₅ photocatalyst for photocatalytic non-oxidative methane coupling. The topic is quite important and this work is timely. The fact of discovering the highly active photocatalyst for this reaction is valuable enough for publication. The phase transition and lattice distortion were found and the distorted sample having the structural defects exhibited high activity, which is an important fact and helpful for the related researchers. The idea of the FLP active sites and activation mechanism seems possible. However, the discussion is unclear and too speculative with less qualitatively, which is not based on the presented figures but somewhat their imaginations. The present manuscript seems not completed and the quality is not suitable for this journal. Thus, the present manuscript is not recommendable for publication in this journal. Comments are following:

1. The citations seem not adequate. They ignore the pioneering and related works of photocatalytic non-oxidative methane coupling (NOCM).
2. The authors mention the methane conversion rate (micromole per gram per hour). But this is not a real reaction rate. To show a real reaction rate, they must use a flow-type reactor instead of a batch reactor as reported (<https://doi.org/10.1039/D0GC01608J>; <https://doi.org/10.1021/acscatal.1c03786>).
3. In addition, the comparison of the photocatalytic performance with “per gram of catalyst” is not suitable.
4. In Fig. 1a, 1f, 3a, 3d, S1, S3, S4, S10b, S15, S17, and S18, the authors might shift the base line (zero line) for each spectrum. Show them quantitatively in a scientific way.
5. The results of ESR and XPS confused the reviewer (Fig. 1 c-d). Explain them quantitatively.
6. The authors mentioned “Sample 873K-Nb₂O₅ exhibits impressive stability when the reaction time is prolonged to 20 h, where the yields of C₂H₆ and H₂ are proportional to the reaction time”. However, the increment decreased with time, which is not “proportional”. In addition, show the balance of hydrocarbons and hydrogen, which was not achieved. This indicates that the undesirable dehydrogenation of hydrocarbons took place and the reaction selectivity of NOCM decreased. Show time course of the reaction selectivity for NOCM.
7. In the case of the reaction experiment for a long-time reaction, we need information of conversion(%). Show conversion(%) for the results of Fig. 2c, S8 and S9.
8. The caption does not fit the data in Fig. S8.
9. The authors mentioned “The 873K-Nb₂O₅ also presents excellent cycling stability, with the negligible variation of ethane production for 4 cycles”. However, this is different from the fact. The balance of hydrogen and ethane increased with the times in the recycling test (Fig. S9). This suggests that the small amount of carbon deposition produced through a side reaction efficiently changed the surface property. Explain this fact.
10. The calculation of the bandgap shown in Fig. 11(Should be S11) in Supplementary information is not acceptable due to the intense absorption by the defects. We cannot know the exact absorption edge in such a case.
11. In Fig. S12, we can see some peaks. The excitation wavelength was 350 nm, which is absorbed by the bulk of Nb₂O₃ samples. What is the origin of the vibrational peaks? It looks like some organic

compounds as impurity adsorbed on the surface. Did the authors clean up the surface with pretreatment such as heating to remove such possible organic impurity?

12. In Fig. S14, although the long-life emission can be observed for some samples, the authors skipped this fact. Explain it.

13. In Fig. S14, the decay curve strangely waved. Why? Confirm the measurement accuracy.

14. This sentence is unclear: "NH₃-temperature-programmed desorption (TPD) reveals the acidity of the calcination temperature for NaBH₄-treated Nb₂O₅ (Fig. S15)."

15. In Fig. S15, the reduced samples showed similar TPD profiles. Are there some differences? Explain more details qualitatively and quantitatively.

16. In Fig. 2f, it is clear that the adsorbed pyridine and pyrrole decreased the photocatalytic activity. However, there is no quantitative explanation and discussion about the number of the acid sites and adsorbed molecules. At present the reviewer cannot understand the speculative conclusion here.

17. In Fig. S16, the variation of the absorption edge with the reduction temperature is unclear. Show them more clearly. Expand the edge region.

18. The authors mentioned "To explicitly understand the cooperation mechanism between acidic and basic sites during methane activation, X-ray absorption spectroscopy (XAS) was further used to finely explore the local structure of reduced Nb₂O₅". Since the specific BET surface area of the samples were small and thus the ratio of the surface Nb atoms to the bulk Nb atoms in the crystals should be small (Fig. S2), XAS spectroscopy is basically not suitable for discussion of the surface acid and base sites. In addition, since "Niobium oxide has a highly asymmetric crystal structure and a wide range of Nb-O bond lengths", the EXAFS analysis has less reliability.

19. The relation between the results of EXAFS analysis and the structural model used for DFT calculation is unclear. Explain the assignment of the shorter Nb-O bonds and longer Nb-O bonds in the model

20. This reviewer is not good at theoretical calculation and thus cannot judge the accuracy of the part, especially the effect of photoirradiation. However, such small cluster model might not be suitable to consider the surface structure of the solid materials.

Reviewer #2 (Remarks to the Author):

The photocatalytic methane conversion efficiency by sole semiconductor has been very low, especially for non-oxidative methane coupling reaction. The idea of construction of high-density FLP sites by promoted phase transition of lamellar Nb₂O₅ for improving the catalytic activity of non-oxidative methane coupling is attractive. The results are discussed in depth and the conclusions are very relevant to the scientific community. Therefore, the manuscript may be acceptable for publication in Nature Communications. Nevertheless some points need to be clarified. For all the above mentioned reason I recommend a minor revision for this manuscript.

1. The authors show that the sample reduced by sodium borohydride has improved absorption of visible light. For such a significant increased light absorption intensity, it is recommended to supplement the

activity under simulated sunlight or pure visible light.

2. Some basic experimental descriptions are missing, such as light intensity, reaction temperature, effective working wavelength range, etc. It is recommended to supplement relevant information in the revised manuscript.

3. The authors explored the effect of phase transition extent on the formation of FLP, and the activation effect of FLP on methane is emphasized in the mechanism discussion section. Is the FLP more important for methane activation or the defects created by the phase transition?

4. It should be pointed out that besides the coupled LA and LB mechanism proposed by the authors, the contribution from the Nb-OH group alone should not be negligible, which may generate active hydroxyl radicals to activate methane. This had been demonstrated to some extent by the quenching experiments using pyridine and pyrrole, respectively. It can be seen that after the neutralization of acidic sites, the catalyst still maintained nearly half of the original activity; however, when the basic sites are neutralized, the activity was greatly reduced.

5. Fig 3a exhibited the variation of Nb-O bond distance with the reduction temperature. Except for the change of the main peak, it can be noted that the 773K-Nb₂O₅ begin to show the broad peak around 2-3 Å. What does the peak change in this range mean?

6. The schematic lines drawn by the author in Fig 1b do not seem to be precisely aligned with the lattice fringes. Please check it carefully.

7. The methane conversion rate is used as the standard for activity comparison in the article. However, it can be seen from Figure 2a that the hydrogen yield from sample 873K-Nb₂O₅ becomes higher, so the selectivity to different alkane products should be supplemented.

8. Fig 2a shows the UV-vis DRS spectra of different Nb₂O₅ samples. While the absorption of visible light is enhanced, the absorption of ultraviolet light is weakened on the NaBH₄-treated samples. For this sample, if the AQY results from the ultraviolet and visible light irradiation can be provided, the contribution of different wavelengths can be better understood.

Reviewer #3 (Remarks to the Author):

The authors studied the non-oxidative coupling of methane (NOCM) over several Nb₂O₅ based materials and found that 873K-Nb₂O₅ sample has high photocatalytic activity of 1,456 μmol/g.h. Based on several spectroscopic observation and theoretical calculations, they claimed that frustrated Lewis pairs consisting of low-valence Lewis acid (LA) and Lewis base (LB) play key roles in the enhancement of photocatalytic NOCM. While interesting, there are three problems in the manuscript at the present

stage.

1. The experimental data presented here do not show any evidence that methane can interact strongly with these surfaces with frustrated Lewis pairs and vacancies. Although IR measurement of Py and TPD of NH₃ shows the existence of frustrated Lewis pairs on their samples, there is no direct experimental evidence that the methane, one of the extremely weakly adsorbed molecular species, does indeed interact with these surface sites and show polarization. If methane molecules strongly interact with their sample surface as proposed in their calculation, then, adsorbed methane is clearly detected by the infrared absorption spectroscopy. It was reported that C-H stretching mode of methane derived from the weakened C-H bonds typically redshifts from the corresponding gas-phase values by ~200 cm⁻¹ [Table 1 of Catal Today 160, 213 (2011)]. Therefore, the existence of the softened C-H vibrational peak of the strongly adsorbed methane species should be demonstrated by IR spectroscopy for these samples under both dark and light irradiation conditions, and the correlation with reaction activity and consistency with the mechanism proposed by their theoretical calculation should be verified carefully. Based on these additional results and arguments, authors should validate their claims and mechanism.
2. Related to the above issue, differences in the interaction of each sample with methane should be directly evaluated as differences in methane adsorption energy by adsorption isotherm and/or TPD measurements [J.Chem.Phys.132,024709(2010)]. It is highly desirable that these basic data is discussed in relation to the photocatalytic effects of the frustrated Lewis pairs and vacancies for each sample.
3. Although authors would want to emphasize methane conversion under mild conditions as in line 88, they did not indicate and discuss the effect of temperature rise and methane pressure under the irradiation of 300 W Xe lamp. In general, sample temperatures substantially rise under the irradiation of such intense light. Can the authors really claim that the reaction is occurring under mild conditions?

These basic issues must be resolved before making an acceptance decision.

I also would like to point out minor problems, as follows.

1. TEM image (Fig. 1b) is blurred and unclear.
2. Most of the figure texts are also blurred and not clear

Reply to the comments of Reviewers

Response to Reviewer 1

General comments: This manuscript shows a discovery of a new Nb₂O₅ photocatalyst for photocatalytic non-oxidative methane coupling. The topic is quite important and this work is timely. The fact of discovering the highly active photocatalyst for this reaction is valuable enough for publication. The phase transition and lattice distortion were found and the distorted sample having the structural defects exhibited high activity, which is an important fact and helpful for the related researchers. The idea of the FLP active sites and activation mechanism seems possible. However, the discussion is unclear and too speculative with less qualitatively, which is not based on the presented figures but somewhat their imaginations. The present manuscript seems not completed and the quality is not suitable for this journal. Thus, the present manuscript is not recommendable for publication in this journal.

Reply: We would like to thank the reviewer for the careful reading of our manuscript. According to your suggestion, we have quantitatively evaluated the acidity content, Vo and hydroxyl groups based on NH₄-TPD and XPS results. The CH₄-TPD was also supplemented to more explicitly verify the effect of FLP sites on methane activation. The *in situ* infrared spectroscopy was further used to corroborate the methane activation by FLP as characterized by the softened C-H vibration mode over FLP-containing Nb₂O₅. The corresponding revisions and supplementary data are highlighted in the revised manuscript and shown as below. We hope our revision can make the paper more acceptable for publication in *Nature Communications*.

Comment 1: The citations seem not adequate. They ignore the pioneering and related works of photocatalytic non-oxidative methane coupling (NOCM).

Reply: We thank the reviewer for this suggestion, enabling us to improve our manuscript. There are many interesting pioneer studies for photocatalytic NOCM reaction, and we have supplemented some related references to give readers a more comprehensive background introduction.

It has been modified in manuscript:

6. Shimura K, Kawai H, Yoshida T, Yoshida H. Bifunctional rhodium cocatalysts for photocatalytic steam reforming of methane over alkaline titanate. *ACS Catalysis* **2**, 2126-2134 (2012).

7. Yoshida H, *et al.* Hydrogen Production from Methane and Water on Platinum Loaded Titanium Oxide Photocatalysts. *The Journal of Physical Chemistry C* **112**, 5542-5551 (2008).

8. Shimura K, Kawai H, Yoshida T, Yoshida H. Simultaneously photodeposited rhodium metal and oxide nanoparticles promoting photocatalytic hydrogen production. *Chemical Communications* **47**, 8958-8960 (2011).

9. Sastre F, Fornés V, Corma A, García H. Selective, room-temperature transformation of methane to C1 oxygenates by deep UV photolysis over zeolites. *Journal of the American Chemical Society* **133**, 17257-17261 (2011).

10. Shoji S, *et al.* Photocatalytic uphill conversion of natural gas beyond the limitation of thermal reaction systems. *Nature Catalysis* **3**, 148-153 (2020).

11. Shimura K, Yoshida H. Hydrogen production from water and methane over Pt-loaded calcium titanate photocatalyst. *Energy & Environmental Science* **3**, 615-617 (2010).

12. Yuliati L, Hamajima T, Hattori T, Yoshida H. Nonoxidative coupling of methane over

- supported ceria photocatalysts. *The Journal of Physical Chemistry C* **112**, 7223-7232 (2008).
21. Lang J, Ma Y, Wu X, Jiang Y, Hu YH. Highly efficient light-driven methane coupling under ambient conditions based on an integrated design of a photocatalytic system. *Green Chemistry* **22**, 4669-4675 (2020).
22. Singh SP, Yamamoto A, Fudo E, Tanaka A, Kominami H, Yoshida H. A Pd-Bi Dual-Cocatalyst-Loaded Gallium Oxide Photocatalyst for Selective and Stable Nonoxidative Coupling of Methane. *ACS Catalysis* **11**, 13768-13781 (2021).
50. Ferrari AM, Huber S, Knözinger H, Neyman KM, Rösch N. FTIR Spectroscopic and Density Functional Model Cluster Studies of Methane Adsorption on MgO. *The Journal of Physical Chemistry B* **102**, 4548-4555 (1998).
51. Li C, Li G, Xin Q. FT-IR spectroscopic studies of methane adsorption on magnesium oxide. *The Journal of Physical Chemistry* **98**, 1933-1938 (1994).
52. Weaver JF, Hinojosa JA, Hakanoglu C, Antony A, Hawkins JM, Asthagiri A. Precursor-mediated dissociation of n-butane on a PdO(101) thin film. *Catalysis Today* **160**, 213-227 (2011).
53. Koitaya T, Ishikawa A, Yoshimoto S, Yoshinobu J. C-H Bond Activation of Methane through Electronic Interaction with Pd (110). *The Journal of Physical Chemistry C* **125**, 1368-1377 (2021).
54. Weaver JF, Hinojosa Jr JA, Hakanoglu C, Antony A, Hawkins JM, Asthagiri A. Precursor-mediated dissociation of n-butane on a PdO (1 0 1) thin film. *Catalysis today* **160**, 213-227 (2011).

Comment 2: The authors mention the methane conversion rate (micromole per gram per hour). But this is not a real reaction rate. To show a real reaction rate, they must use a flow-type reactor instead of a batch reactor as reported. (<https://doi.org/10.1039/D0GC01608J>; <https://doi.org/10.1021/acscatal.1c03786>)

Reply: We thank the reviewer's comment and agree the flow-type reactor is more appropriate for evaluating the reaction rate. According to your requirement, we have carried out the corresponding experiments based on flow-type reactor. The References(<https://doi.org/10.1039/D0GC01608J>;<https://doi.org/10.1021/acscatal.1c03786>) have been cited in our revised manuscript. The reaction conditions are as follows: The catalyst powder was pressed under 40 MPa pressure and ground into 40–60 mesh. Photocatalyst in a quartz cell, 0.05 g; photoirradiation area, 28.26 mm²; cell volume, 113.04 mm³; feed gas, 99.999% of CH₄ in flow rate 10 mL min⁻¹; SV:55000 h⁻¹; light intensity, 2000 mW cm⁻².

Fig. R1 The reaction device (left) and the activity data obtained from flow reactor (right).

The corresponding reaction device and activity data are provided in Fig. R1. Since the conditions of the flow-type reactor have higher requirements on the activity of the catalyst, the catalysts with relatively low activity and after active site quenching cannot be accurately evaluated in the flow-type reactor. To more thoroughly compare the activity with the current reports (Table S4) and accurately demonstrate the proposed mechanism, we have presented the activity data both from the batch and flow reactors in the revised manuscript.

Comment 3: In addition, the comparison of the photocatalytic performance with “per gram of catalyst” is not suitable.

Reply: Thank for your suggestion. We corrected the way of activity expression, and provided AQY to more thoroughly evaluate the photocatalytic activity of FLP-containing Nb₂O₅.

Fig R2. The photocatalytic activity of 873K-Nb₂O₅ for the NOCM reaction in a flow reactor. Reaction conditions: 0.05 g catalyst; photoirradiation area, 28.26 mm²; feed gas, 99.999% of CH₄ in flow rate 10 mL min⁻¹; SV:55000 h⁻¹; irradiated under 300 W Xe lamp for 60 h.

It has been modified in manuscript:

To further demonstrate the ability of catalysts to activate methane, the NOCM reaction in a flow reactor was carried out (Fig 2b, Fig S11 and Table S8). After a quick induction period, the 873K-Nb₂O₅ has a stable ethane production rate of around 22.52 μmol h⁻¹ without significant decrease within 60 h. The ethane and hydrogen show a stable equimolar ratio during the reaction. The apparent quantum yield (AQY) was further calculated to demonstrate the methane conversion capability of this catalyst (Table S9). The AQY of 873K-Nb₂O₅ for methane conversion is 0.43% under 365 nm light irradiation and significantly decreases under the visible light irradiation.

Comment 4: In Fig. 1a, 1f, 3a, 3d, S1, S3, S4, S10b, S15, S17, and S18, the authors might shift the base line (zero line) for each spectrum. Show them quantitatively in a scientific way.

Reply: We thank the reviewer for pointing this problem out. The pictures mentioned by the reviewers have all been revised. In partial figure, if the lines of different samples are superimposed, it will not be possible to distinguish the details. The scientific scales were added on the y-axis as a reference for better representation. The original Fig 3d and S18 exhibit the upper and lower parts of the Nb K-edge EXAFS (points) and curvefit (line) for 873K-Nb₂O₅ are FT magnitude and imaginary components in k³-weighted R-space, respectively, and moving

the baseline has no effect on the results.

It has been modified in manuscript:

Fig 1a

Fig 1f

Fig 3a

Fig S1

Fig S3

original S4

original Fig S10b

original Fig S15

original Fig S17

Comment 5: The results of ESR and XPS confused the reviewer (Fig. 1 c-d). Explanation them quantitatively.

Reply: We have revised the corresponding description in the manuscript as follows. We hope the revision has well explained the situation about Vo.

It has been modified in manuscript:

The refined elements X-ray photoelectron spectroscopy (XPS) and electron paramagnetic

resonance (EPR) were further used for analyzing the effect of thermal reduction on the electronic state of Nb₂O₅. The refined O 1s XPS spectra of 873K-Nb₂O₅ exhibits much higher peak at 531.3 eV corresponding to oxygen vacancy (Vo) sites than that of H₂-treated Nb₂O₅ at the same temperature (Fig 1c and d, Fig S4). It is also noted the peak of hydroxyl groups centered at 532.0 eV is significantly improved for 873K-Nb₂O₅, which is confirmed by the FTIR analysis as characterized by the broad stretching vibration bands around 3415 cm⁻¹ and bending vibration bands around 1648 cm⁻¹ (Fig 1e). The XPS peak of hydroxyl groups decreases with the further increasing of calcination temperature (973K-Nb₂O₅, Fig S5). Together with the XRD results, the structure distortion and atomic dislocation of sample 873K-Nb₂O₅ from the low-temperature phase transition should result in the simultaneous formation of surface Vo and hydroxyl groups. On the other hand, the EPR analysis indicates that although the signal at g = 2.004 attributed to Vo is improved from the thermal reduction compared with Nb₂O₅ (Fig S5, Fig 1f), sample 873K-Nb₂O₅ with the most distorted crystal structure exhibits a lower signal of Vo compared with H₂-treated counterpart (Fig 1c). It is known EPR is only applicable to the detection of single-electron occupied species, so the discrepancy between the EPR and XPS analyses regarding the Vo content should be ascribed to the formation of less single-electron occupied Vo in 873K-Nb₂O₅. The most distorted sample of 873K-Nb₂O₅ has the highest contents of Vo and hydroxyl groups according to the XPS and FTIR (Fig S4, Fig 1e), suggesting that the formation of less single-electron occupied Vo in 873K-Nb₂O₅ should be related to the concomitant formation of hydroxyl groups. Moreover, the refined Nb 3d XPS spectra indicate 873K-Nb₂O₅ has a higher content of low-valence Nb (Nb_{LV}) than that of H₂-treated Nb₂O₅, which should be attributed to the Nb neighboring Vo (Fig S6). It is thus postulated that hydroxyl groups should help promote the electron delocalization from Vo to neighboring Nb, forming more low-valent Nb in sample 873K-Nb₂O₅. The above speculation about the interaction with Vo and hydroxyl groups will be corroborated later by combining with other analytic techniques and DFT calculations.

Comment 6: The authors mentioned “Sample 873K-Nb₂O₅ exhibits impressive stability when the reaction time is prolonged to 20 h, where the yields of C₂H₆ and H₂ are proportional to the reaction time”. However, the increment decreased with time, which is not “proportional”. In addition, show the balance of hydrocarbons and hydrogen, which was not achieved. This indicates that the undesirable dehydrogenation of hydrocarbons took place and the reaction selectivity of NOCM decreased. Show time course of the reaction selectivity for NOCM.

Reply: We thank the reviewer for pointing this problem out. When carrying out the long-term reaction experiment of the original batch reactor, the gas production can only be continuously extracted from the same 44.6 mL reactor for detection limited by the conditions. Three sampling are taken every 4 hours as a time point, and supplemented with inert gas to atmospheric pressure. This results in a decrease in methane gas inside the reactor over time. By the time of reaction for 20 hours, theoretically, the amount of methane gas reacted was reduced by 15 mL compared with the initial reaction condition. Converted according to this ratio, it can be seen that the increase of the main product of 873K-Nb₂O₅ in 20 hours is proportional to the time. No conversions have been performed in order to show actual measured data. We supplemented the 60-hour experiment in a flow-type reactor according to the reviewer's comments, from which it can also be concluded that the ethane and hydrogen were produced stably and continuously.

To remove ambiguity, we have revised the description of the 20 h reaction in batch reactor in the original text.

For the proportional balance of hydrocarbons and hydrogen products, this is mainly caused by the detection limit of the gas chromatography and batch reactor. The gas chromatography in our lab can only detect products with 4 carbons, but in all 4-hour activity comparison experiments, 20-hour long-term reaction experiments and 4 rounds of cycle experiments, we have detected more hydrogen than alkane products from 873K-Nb₂O₅. In order to explore the composition of the product more accurately, we performed a combined gas chromatography-mass spectrometry detection of the product in the batch reactor for 4 hours. It can be seen that through mass spectrometry and detection with higher sensitivity, during the continuous reaction in the batch reactor, the main product will undergo further coupling to produce multi-carbon alkanes. This process leads to a large increase in the detected hydrogen. This is also an unavoidable deep side reaction of batch reactors. Our detection in the flow reactor benefits from the rapid discharge of the product, and only detects ethane and hydrogen in a molar ratio. Taking into account the further increase of multi-carbon products in the 20-hour reaction and the reduction of reaction gas mentioned above, the relationship between the ratio of hydrogen to alkanes will not be discussed here.

PK#	RT	Area%	LibraryID	Ref#	CAS#	Qual
1	1.836	38.37	D:\MSDCHEM\Library\NIST20.L			
Methane & Ethane & Propane						
2	2.009	48.57	D:\MSDCHEM\Library\NIST20.L			
Butane						
3	2.232	1.17	D:\MSDCHEM\Library\NIST20.L			
Butane, 2-methyl						
4	2.353	0.54	D:\MSDCHEM\Library\NIST20.L			
Pentane						
5	2.598	0.05	D:\MSDCHEM\Library\NIST20.L			
Butane, 2,2-dimethyl						
6	2.900	0.07	D:\MSDCHEM\Library\NIST20.L			
Pentane, 2-methyl						
7	3.079	9.17	D:\MSDCHEM\Library\NIST20.L			
Pentane, 3-methyl						
8	3.306	0.73	D:\MSDCHEM\Library\NIST20.L			
n-Hexane						
9	4.464	0.14	D:\MSDCHEM\Library\NIST20.L			
Pentane, 3,3-dimethyl						
10	4.853	0.05	D:\MSDCHEM\Library\NIST20.L			
Pentane, 2,3-dimethyl						
11	5.049	0.27	D:\MSDCHEM\Library\NIST20.L			
Hexane, 3-methyl						
12	8.037	0.48	D:\MSDCHEM\Library\NIST20.L			
Hexane, 3,4-dimethyl & Pentane, 3-ethyl-3-methyl						
13	8.121	0.08	D:\MSDCHEM\Library\NIST20.L			
Hexane, 2,4-dimethyl						

It has been modified in manuscript:

The 873K-Nb₂O₅ exhibits stability in the batch reactor when the reaction time is prolonged to 20 h, where the yields of C₂H₆ and H₂ are continuous and stable (Table S5). In stark contrast, methane could not be continuously activated after 8 h over pristine and H₂-treated Nb₂O₅ (Fig S8 and Fig S9). After 20 hours, the conversion rates of CH₄ over pristine, H₂-treated Nb₂O₅ and 873K-Nb₂O₅ are 0.1%, 1.2%, and 4.3%. The 873K-Nb₂O₅ also presents good cycling stability, with slightly decrease in the yields of ethane and hydrogen, which is possibly due to a small amount of carbon deposition (Fig S10 and Table S6).

To further demonstrate the ability of catalysts to activate methane, the NOCM reaction in a flow reactor was carried out (Fig 2b, Fig S11 and Table S7). After a quick induction period, the 873K-Nb₂O₅ has a stable ethane production rate of around 22.52 μmol h⁻¹ without significant decrease within 60 h. The ethane and hydrogen show a stable equimolar ratio during the reaction.

Comment 7: In the case of the reaction experiment for a long-time reaction, we need information of conversion(%). Show conversion(%) for the results of Fig. 2c, S8 and S9.

Reply: Thanks for your suggestion. We have provided the conversion rate of different samples according to your suggestion.

It has been modified in manuscript:

The 873K-Nb₂O₅ exhibits stability in the batch reactor when the reaction time is prolonged to 20 h, where the yields of C₂H₆ and H₂ are continuous and stable (Table S5). In stark contrast, methane could not be continuously activated after 8 h over pristine and H₂-treated Nb₂O₅ (Fig S8 and Fig S9). After 20 hours, the conversion rates of CH₄ over pristine, H₂-treated Nb₂O₅ and 873K-Nb₂O₅ are 0.1%, 1.2%, and 4.3%.

Comment 8: The caption does not fit the data in Fig. S8.

Reply: Thanks for your kind reminding, this problem has been corrected.

It has been modified in manuscript:

Fig S8. The long time photocatalytic NOCM reactions over (a) Nb₂O₅, (b) H₂-treated Nb₂O₅, and (c) 873K-Nb₂O₅.

Comment 9: The authors mentioned “The 873K-Nb₂O₅ also presents excellent cycling stability, with the negligible variation of ethane production for 4 cycles”. However, this is different from the fact. The balance of hydrogen and ethane increased with the times in the recycling test (Fig. S9). This suggests that the small amount of carbon deposition produced through a side reaction efficiently changed the surface property. Explain this fact.

Reply: Thanks for reminding us of this point. The hydrogen in excess of ethane stoichiometric ratio was derived from further coupling of the ethane product as verified by GC-MS. In the cycle experiment, it can be seen that the yield of ethane remains basically unchanged, while the yield of hydrogen decreases. The carbon residual is only slightly varied during cyclic reaction. The long-time NOCM reaction in the flow reactor shows a stable equimolar ratio of ethane and hydrogen. The slight change in the balance between hydrogen and ethane should be caused by the gradual accumulation of undesorbed products during the cyclic batch reaction, resulting in some measurement errors, and a small amount of carbon deposition may also have a certain impact.

It has been modified in manuscript:

The 873K-Nb₂O₅ also presents good cycling stability, with slightly decrease in the yields of ethane and hydrogen, which is possibly due to a small amount of carbon deposition (Fig S10 and Table S6).

To further demonstrate the ability of catalysts to activate methane, the NOCM reaction in a flow reactor was carried out (Fig 2b, Fig S11 and Table S7). After a quick induction period, the 873K-Nb₂O₅ has a stable ethane production rate of around 22.52 μmol h⁻¹ without significant decrease within 60 h. The ethane and hydrogen show a stable equimolar ratio during the reaction.

Elemental analysis (EA) was used to investigate carbon deposits on different samples. Table S9 displays the residual coke over different samples. Negligible carbon residual is observed from sample 873K-Nb₂O₅, which only increases from 0.6% to 0.76%. The slightly varied balance between hydrogen and ethane should be caused by the gradually accumulated undesorbed product during cyclic batch reaction. Severe carbon deposition was produced on H₂-treated Nb₂O₅, which demonstrated that the Vo site alone does not efficiently conduct the

process of methane coupling.

Comment 10: The calculation of the bandgap shown in Fig. 11(Should be S11) in Supplementary information is not acceptable due to the intense absorption by the defects. We cannot know the exact absorption edge in such a case.

Reply: Thanks for reminding us of this important point. The comments made by the reviewers are more appropriate and we have removed the image.

Comment 11: In Fig. S12, we can see some peaks. The excitation wavelength was 350 nm, which is absorbed by the bulk of Nb₂O₅ samples. What is the origin of the vibrational peaks? It looks like some organic compounds as impurity adsorbed on the surface. Did the authors clean up the surface with pretreatment such as heating to remove such possible organic impurity?

Reply: We thank the reviewer for this suggestion. The samples avoid the participation of organic matter in the preparation and testing process. Similar results were obtained for the same series of samples on another instrument. These peaks were assigned in article SI.

It has been modified in SI:

The broad peak observed centered at 370 nm corresponds to the pristine band-edge emission of Nb₂O₅. The other complex bands obtained in range of 410– 490 nm are assigned to the deep level emission, respectively, belonging to the emission of band edge free excitons, bound excitons (400–425 nm), surface defects (450–480 nm), and structural defects such as distorted NbO₆ octahedral groups (490 nm).^{27,28}

Comment 12: In Fig. S14, although the long-life emission can be observed for some samples, the authors skipped this fact. Explain it.

Reply: We thank the reviewer for this suggestion. The detailed description of this graph is supplemented in the article SI.

It has been modified in SI:

The values of the lifetime constants (τ) are shown in Table S10. The PL peak decay of NaBH₄-treated-Nb₂O₅ was slower than that of pristine and H₂-treated Nb₂O₅. The 873K-Nb₂O₅ with the longest lifetime of 3.448 ns compared to 0.066 ns of pristine Nb₂O₅, reveals a long life of photogenerated electrons in the excited state, which is highly desirable for the surface reaction. For 873K-Nb₂O₅, the double-exponential model suggested that two emissive states were involved in the PL decay with the fast decay component τ_1 and the much slower component τ_2 , revealing that oxygen vacancies and various defects generated during phase transition effectively inhibit electron/hole recombination.

Comment 13: In Fig. S14, the decay curve strangely waved. Why? Confirm the measurement accuracy.

Reply: Thanks to the reviewers for your comments. Adjusting the frequency can make these baselines close to the same level, and it will not affect the results if they are not at the same level.

Comment 14: This sentence is unclear: “NH₃-temperature-programmed desorption (TPD) reveals the acidity of the calcination temperature for NaBH₄-treated Nb₂O₅.”

Reply: We thank the reviewer for pointing this out, enabling us to improve our manuscript. The detailed analysis of this characterization is supplemented in the article.

It has been modified in manuscript:

NH₃-TPD was further used to reveal the effect of calcination temperature on the acidic characteristics of NaBH₄-treated Nb₂O₅ (Fig 3d). The pristine T-Nb₂O₅ has negligible acid sites for NH₃ adsorption, while all samples treated with NaBH₄ show two distinct peaks above 600 °C attributed to strong acid sites. The H₂-Nb₂O₅ only shows a less intense peak at the lower temperature, suggesting a weaker acidity. The relation between the reduction temperature and acidity is confirmed by pyridine-FTIR (Fig 3e, Table S14). Through peak fitting of NH₃-TPD spectra (Table S15), it seems that sample 873K-Nb₂O₅ with the most distorted structure has the highest concentration of acid sites. It is also noted the acidity trend of NaBH₄-treated Nb₂O₅ is accordant with the change of V_o concentration. NH₃-TPD was also used to verify the quenching of acid sites through pyridine treatment. The results proved that the strongly acidic sites were indeed covered since the adsorption capacity of the sample for ammonia decreases significantly (Fig S20), which is accordant with the CH₄-TPD results.

Fig 3d,original Fig S15

Comment 15: In Fig. S15, the reduced samples showed similar TPD profiles. Are there some differences? Explain more details qualitatively and quantitatively.

Reply: We thank the reviewer for pointing this problem out, enabling us to improve our manuscript. The detailed analysis of this characterization is supplemented in the article. The specific content is replied in comment 14.

Comment 16: In Fig. 2f, it is clear that the adsorbed pyridine and pyrrole decreased the photocatalytic activity. However, there is no quantitative explanation and discussion about the number of the acid sites and adsorbed molecules. At present the reviewer cannot understand the speculative conclusion here.

Reply: We thank the reviewer for pointing this problem out, enabling us to improve our manuscript. The specific contents of acid sites was provided based on NH₃-TPD analysis. The NH₃ desorption peak of the sample treated by pyridine disappeared according to NH₃-temperature-programmed desorption analysis (NH₃-TPD), demonstrating the quenching of acid sites by pyridine. The CH₄-TPD measurement indicates that the desorption temperature of 873K-Nb₂O₅ for CH₄ molecules is decreased after treated by pyridine, suggesting the decreased affinity of 873K-Nb₂O₅ to CH₄ molecules due to the shielding of acid sites by pyridine. These results demonstrated the decreased photocatalytic activity is ascribed to the decreased

chemical adsorption of CH₄ due to the quenching of the acid sites. The detailed analysis of this characterization is supplemented in the article.

It has been modified in manuscript:

Page 8: To verify the effect of site quenching, CH₄-temperature-programmed desorption (TPD) measurement evaluated the differences in methane adsorption,⁴⁴ and explained the affinity of acid/base site toward methane molecule and the trend of catalytic activities (Fig 2f and Table S12). The broad desorption bands can be observed in the CH₄-TPD profiles of different catalysts. The wide gas desorption peaks at middle temperature range (200-400 °C) correspond to chemical adsorption. Except for 873K-Nb₂O₅, most of the methane desorbs in a main peak around 250 °C, and there is a trace amount desorbing around 330 °C. While the chemical desorption peak area in higher temperature range increases significantly as compared with 873K-Nb₂O₅, indicating that more CH₄ is firmly adsorbed on the surface with FLP sites by chemisorption. The additional peak around 400 °C indicates double sites can further enhance the CH₄ adsorption capacity through increasing the binding strength for CH₄. The desorption yield of methane of 873K-Nb₂O₅ is evident higher than single Vo site and 873K-Nb₂O₅ after acid/base quenched. This trend of the amount of adsorbed methane was well matched with the trend of the methane activation after site quench showed in Fig 2e, suggesting that methane adsorption behavior should be closely related to the methane activation.

Page 10: NH₃-TPD was further used to reveal the effect of calcination temperature on the acidic characteristics of NaBH₄-treated Nb₂O₅ (Fig 3d). The pristine T-Nb₂O₅ has negligible acid sites for NH₃ adsorption, while all samples treated with NaBH₄ show two distinct peaks above 600 °C attributed to strong acid sites. The H₂-Nb₂O₅ only shows a less intense peak at the lower temperature, suggesting a weaker acidity. The relation between the reduction temperature and acidity is confirmed by pyridine-FTIR (Fig 3e, Table S14). Through peak fitting of NH₃-TPD spectra (Table S15), it seems that sample 873K-Nb₂O₅ with the most distorted structure has the highest concentration of acid sites. It is also noted the acidity trend of NaBH₄-treated Nb₂O₅ is accordant with the change of Vo concentration. NH₃-TPD was also used to verify the quenching of acid sites through pyridine treatment. The results proved that the strongly acidic sites were indeed covered since the adsorption capacity of the sample for ammonia decreases significantly (Fig S20), which is accordant with the CH₄-TPD results.

Comment 17: In Fig. S16, the variation of the absorption edge with the reduction temperature is unclear. Show them more clearly. Expand the edge region.

Reply: We thank the reviewer's suggestion, relevant diagrams have been added as requested.

It has been modified in manuscript:

Original Fig S17. Normalized Nb K-edge XANES spectra of different Nb₂O₅ samples. The position of the dotted line in b is the highest point of corresponding first derivative absorption curves of Nb₂O₅.

Comment 18: The authors mentioned “To explicitly understand the cooperation mechanism between acidic and basic sites during methane activation, X-ray absorption spectroscopy (XAS) was further used to finely explore the local structure of reduced Nb₂O₅”. Since the specific BET surface area of the samples were small and thus the ratio of the surface Nb atoms to the bulk Nb atoms in the crystals should be small (Fig. S2), XAS spectroscopy is basically not suitable for discussion of the surface acid and base sites. In addition, since “Niobium oxide has a highly asymmetric crystal structure and a wide range of Nb-O bond lengths”, the EXAFS analysis has less reliability.

Reply: We thank the reviewer for pointing this problem out, enabling us to improve our manuscript. XAS analysis was used to assist in the analysis of the material structure. For crystalline materials with a very complex structure such as mixed crystal phase niobium pentoxide, the fitting results are average values, which serve as an auxiliary information for understanding the material structure. By comparing the different degrees of crystal phase transition in XRD and the contribution of different pathways in the fitting analysis, we conclude that the crystal phase transition is not the only reason for the different results in the fitting. We have revised the description about this paragraph according to the reviewer's comments. In order to verify the existence of FLP on the surface of the catalyst, we supplemented the verification of acidic sites, the verification of methane adsorption and the activation of C-H bonds in other comments to further prove the activation of FLP sites on methane. It is also expected the FLP constructed over materials with large specific surface area will further promote the NOCM efficiency. We will continue to explore this possibility in our future study and thank you for your constructive comment.

It has been modified in manuscript:

Page 10: Considering asymmetry and distortion in the crystal structure of NaBH₄-treated Nb₂O₅, and the wide Nb-O distance distribution range, as few paths as possible were chosen to investigate the overall situation of the first coordination shell. The fitting results only represent the average atomic arrangement environment in the sample. Although the fitting results show

a simplified average coordination environment, it is supposed that the effect of the reduction treatment on the average Nb-O path can be reflected from fitting. For the cases of pristine and H₂-treated Nb₂O₅, representative fits can be achieved through one average Nb-O path, while the complex situation of NaBH₄-treated Nb₂O₅ requires one shorter Nb-O path together with a longer Nb-O path (Fig 3c, Fig S18 and Table S13), confirming the lattice distortion from NaBH₄ treatment. To explicitly understand the origin of the longer Nb-O path, the ratio of surface hydroxyl groups is plotted versus the ratio of fitted longer Nb-O path (Fig S19). It is clear that the NaBH₄ treatment results in more significant increase of long Nb-O distance than hydroxyls. Specifically, the percentage of long Nb-O distance increases for about 30% for 673K-Nb₂O₅, 773K-Nb₂O₅ and 973K-Nb₂O₅, while that of the hydroxyl increase for less than 10%, which demonstrates the long Nb-O distance is mainly caused by the structure distortion in these samples. In comparison, 873K-Nb₂O₅ shows a further increase of long Nb-O distance and hydroxyl group with more comparable percentage, which suggest the contribution from hydroxyl to the long-distance Nb-O bond is improved in 873K-Nb₂O₅.

Comment 19: The relation between the results of EXAFS analysis and the structural model used for DFT calculation is unclear. Explain the assignment of the shorter Nb-O bonds and longer Nb-O bonds in the model.

Reply: We thank the reviewer for pointing this problem out, enabling us to improve our manuscript. We understand the doubts of the reviewers. It is well known that EXAFS does not depend on the crystal structure, and it is often used in the study of a large number of amorphous materials and materials with uncertain crystal structures. So we use EXAFS to compare the structure in mixed crystal phase Nb₂O₅. Since the average path of Nb-O rather than the exact Nb-O bond is reflected in the R space, we use fewer paths for fitting to understand the average situation of the Nb-O coordination. The Debye Waller factor can understand the change of disorder in the system. The origin of path extraction will be explained first. Take the example of the structure in which metal atoms are coordinated by oxygen atoms and sulfur atoms at the same time. In EXAFS fitting, the standard oxide of the metal is usually used to extract the M-O path, and the standard sulfide is used to extract the M-S path. This is because the k-space points of the standard path are more orderly. Some researchers also use the similar distance M-O path and M-S extracted from the crystal model established by themselves for fitting. These are feasible operations. When looking for a path, the path with suitable R_{eff} can be used to fit.

We did not use the cluster model in the paper for fitting, this is because the Feff path operation must be performed using the crystal file. We use the cluster structure in the DFT calculation mainly for the consideration of reducing the calculation cost. This cluster structure is sufficient for the calculation of various adsorption and transition states of the FLP site, and is also suitable for subsequent Gaussian calculations. We use the simplified T-Phase crystal model (Figure 4a in article) to extract the path, and perform Feff calculations on multiple different Nb atoms, as shown in the figure below. Our cluster structure is also obtained from this crystal model.

	Degen	Reff	Scattering path	Rank	I	Type
1	1.00	1.987	@ 041.1 @	100.00	↓	single scattering
2	3.00	2.035	@ 073.1 @	100.00	↓	single scattering
3	1.00	2.078	@ 038.1 @	31.53	↓	single scattering
4	1.00	2.139	@ 042.1 @	29.19	↓	single scattering
5	1.00	2.268	@ 09.1 @	24.91	↓	single scattering
6	2.00	3.225	@ 028.1 038.1 @	3.64	↓	acute triangle
7	1.00	3.282	@ Nb34.1 @	12.81	↓	single scattering
8	4.00	3.313	@ 073.1 028.1 @	5.83	↓	other double scat
9	2.00	3.335	@ 05.1 09.1 @	3.18	↓	acute triangle
12	1.00	3.410	@ Nb11.1 @	11.57	↓	single scattering
13	4.00	3.438	@ 041.1 073.1 @	3.84	↓	other double scat
17	1.00	3.498	@ Nb2.1 @	10.81	↓	single scattering
22	4.00	3.751	@ 073.1 042.1 @	3.25	↓	other double scat
23	2.00	3.750	@ Nb108.1 @	17.87	↓	single scattering
24	4.00	3.768	@ 028.1 Nb11.1 @	5.66	↓	obtuse triangle
25	2.00	3.800	@ 05.1 Nb2.1 @	3.35	↓	obtuse triangle
26	1.00	3.853	@ 071.1 @	5.15	↓	single scattering
27	2.00	3.874	@ 028.1 Nb34.1 @	5.45	↓	obtuse triangle

Nb1

	Degen	Reff	Scattering path	Rank	I	Type
1	1.00	1.987	@ 043.1 @	100.00	↓	single scattering
2	4.00	2.072	@ 06.1 @	100.00	↓	single scattering
3	1.00	2.121	@ 0111.1 @	23.48	↓	single scattering
4	1.00	2.209	@ 044.1 @	21.06	↓	single scattering
5	4.00	3.257	@ 05.1 09.1 @	5.43	↓	acute triangle
6	1.00	3.286	@ Nb108.1 @	10.01	↓	single scattering
9	1.00	3.375	@ Nb107.1 @	9.33	↓	single scattering
11	4.00	3.424	@ 043.1 06.1 @	3.24	↓	other double scat
13	1.00	3.498	@ Nb1.1 @	8.47	↓	single scattering
15	1.00	3.512	@ 042.1 @	5.46	↓	single scattering
16	1.00	3.588	@ Nb3.1 @	7.91	↓	single scattering
19	4.00	3.679	@ 0111.1 Nb108.1 @	4.27	↓	other double scat
24	2.00	3.815	@ 06.1 Nb3.1 @	3.25	↓	obtuse triangle
25	1.00	3.824	@ 0145.1 @	4.15	↓	single scattering
26	1.00	3.841	@ Nb34.1 @	6.55	↓	single scattering
27	1.00	3.895	@ 0143.1 @	3.91	↓	single scattering
28	2.00	3.911	@ 09.1 Nb34.1 @	7.11	↓	obtuse triangle
29	2.00	3.940	@ 041.1 @	7.24	↓	single scattering

Nb2

	Degen	Reff	Scattering path	Rank	I	Type
1	1.00	1.900	@ 0113.1 @	100.00	↓	single scattering
2	2.00	1.948	@ 07.1 @	100.00	↓	single scattering
3	1.00	1.981	@ 047.1 @	47.91	↓	single scattering
4	1.00	2.013	@ 035.1 @	45.99	↓	single scattering
5	1.00	2.292	@ 048.1 @	32.73	↓	single scattering
6	2.00	3.150	@ 0113.1 08.1 @	4.46	↓	other double scat
7	1.00	3.180	@ 074.1 @	13.24	↓	single scattering
8	2.00	3.183	@ 07.1 035.1 @	4.49	↓	other double scat
9	2.00	3.284	@ Nb105.1 @	35.21	↓	single scattering
10	2.00	3.308	@ 08.1 047.1 @	3.13	↓	other double scat
12	4.00	3.375	@ 0113.1 07.1 @	5.47	↓	other double scat
13	1.00	3.389	@ 0140.1 @	10.73	↓	single scattering
15	1.00	3.483	@ 0114.1 @	9.87	↓	single scattering
17	2.00	3.588	@ 08.1 035.1 @	3.07	↓	other double scat
20	6.00	3.671	@ 08.1 Nb105.1 @	10.97	↓	other double scat
21	1.00	3.675	@ 046.1 @	8.36	↓	single scattering
24	2.00	3.750	@ Nb31.1 @	24.86	↓	single scattering
25	2.00	3.764	@ 08.1 @	15.50	↓	single scattering

Nb4

It can be seen that for a crystal with a very asymmetric and complex structure such as niobium pentoxide, different Nb atoms in it will have different Feff paths. And the R space data is the averaged Nb-O coordination situation, which is doomed that we can only conduct comparative analysis between different samples from the overall structure change. Let's take prsitine Nb₂O₅ as an example. We choose path 4 in Nb1 Feff and path 3 in Nb2 Feff, and we get similar results. A path can roughly show the situation in the sample. When we take 873K-Nb₂O₅ as an example, we can find that none of the above-mentioned single path can fit well. This is because the 873K-Nb₂O₅ sample has undergone a very large crystal distortion, and a large number of hydroxyl groups have also been produced on the surface. We choose a shorter path (path 2 in Nb4, Reff is 2.013) and a longer path (path 4 in Nb2, Reff is 2.209) to fit again, and we can get better results.

There are two main reasons why two paths are needed to fit NaBH₄-treated Nb₂O₅: 1. The samples treated with NaBH₄ have undergone different degrees of crystal phase transformation, which means that a large amount of distortion, stretching and dislocation. A single Nb-O path is not enough to show the average result, and two paths can better show the disorder of Nb-O coordination that occurs in NaBH₄-treated samples. 2. Because the Nb-OH distance of the surface hydroxyl is relatively long, the resulting Nb-O coordination distance change in the surface hydroxyl is also reflected in the longer Nb-O pathway. We suggest that the surface hydroxyl groups only partially contribute to the longer Nb-O pathway. Considering the proportion of surface Nb atoms is not majority due to the small specific BET surface area, surface hydroxyls should account for a small proportion of the longer Nb-O pathway. From the XRD analysis above, 973K sample has the largest crystal phase transformation degree, but the 873K sample has the largest proportion of the longer path in the fitting summary, which shows that the crystal phase distortion stretching is not the only source of the longer Nb-O path. It also suggest the surface hydroxyl content in 873K-Nb₂O₅ is much higher than that of other NaBH₄-treated Nb₂O₅. The Nb-O bonds in the abundant surface hydroxyls lead to increased average Nb-O distances in the R space. By observing the relationship between the proportion of surface hydroxyl groups and the proportion of longer paths, it can be seen that the 873K sample needs the most long Nb-O path fitting, which is caused by the crystal phase transition and the presence of surface hydroxyl groups.

It has been modified in manuscript:

Page 10: Considering asymmetry and distortion in the crystal structure of NaBH₄-treated Nb₂O₅, and the wide Nb-O distance distribution range, as few paths as possible were chosen to investigate the overall situation of the first coordination shell. The fitting results only represent the average atomic arrangement environment in the sample. Although the fitting results show a simplified average coordination environment, it is supposed that the effect of the reduction treatment on the average Nb-O path can be reflected from fitting. For the cases of pristine and H₂-treated Nb₂O₅, representative fits can be achieved through one average Nb-O path, while the complex situation of NaBH₄-treated Nb₂O₅ requires one shorter Nb-O path together with a longer Nb-O path (Fig 3c, Fig S18 and Table S13), confirming the lattice distortion from NaBH₄ treatment. To explicitly understand the origin of the longer Nb-O path, the ratio of surface hydroxyl groups is plotted versus the ratio of fitted longer Nb-O path (Fig S19). It is clear that the NaBH₄ treatment results in more significant increase of long Nb-O distance than hydroxyls. Specifically, the percentage of long Nb-O distance increases for about 30% for 673K-Nb₂O₅, 773K-Nb₂O₅ and 973K-Nb₂O₅, while that of the hydroxyl increase for less than 10%, which demonstrates the long Nb-O distance is mainly caused by the structure distortion in these samples. In comparison, 873K-Nb₂O₅ shows a further increase of long Nb-O distance and hydroxyl group with more comparable percentage, which suggest the contribution from hydroxyl to the long-distance Nb-O bond is improved in 873K-Nb₂O₅.

Comment 20:This reviewer is not good at theoretical calculation and thus cannot judge the accuracy of the part, especially the effect of photoirradiation. However, such small cluster model might not suitable to consider the surface structure of the solid materials.

Reply: We understand reviewers' concerns. The use of the cluster structure is mainly due to two considerations:1. The crystal structure of T-Nb₂O₅ is too complicated to be displayed in the

article, and the cost required to select the overall crystal form for calculation is too high. We selected clusters to show the activation of methane on FLP sites. Even if the entire unit cell is selected for calculation, it is only a single FLP site with different structures on the huge crystal surface for subsequent calculations, which consumes too much time and cost. There is no difference in the final conclusion from the calculation with the cluster structure. The selected part of the structure has been shown in the article that a variety of different FLP sites can be constructed, which is enough to prove the abundance and effectiveness of FLP on Nb₂O₅. 2. We use Gaussian to calculate the methane activation process under light excitation. This software requires the cluster structure for calculation. Therefore, we use clusters to perform a series of optimization calculations in DFT calculation, and use the same model for further TD-DFT calculation to ensure the consistency and accuracy of the data. Therefore, this cluster structure can be used to discuss and analyze the activation of FLP sites for methane.

Response to Reviewer 2

General comments: The photocatalytic methane conversion efficiency by sole semiconductor has been very low, especially for non-oxidative methane coupling reaction. The idea of construction of high-density FLP sites by promoted phase transition of lamellar Nb₂O₅ for improving the catalytic activity of non-oxidative methane coupling is attractive. The results are discussed in depth and the conclusions are very relevant to the scientific community. Therefore, the manuscript may be acceptable for publication in *Nature Communications*. Nevertheless some points need to be clarified. For all the above mentioned reason I recommend a minor revision for this manuscript.

Reply: We appreciate the reviewer for their generous recommendation of our manuscript for publication in *Nature Communications*. The suggestions and comments have been carefully considered and we made the corresponding revision to address the points the reviewer has raised, as shown below. We hope our revision can make the paper much more acceptable for publication in *Nature Communications*.

Comment 1: The authors show that the sample reduced by sodium borohydride has improved absorption of visible light. For such a significant increased light absorption intensity, it is recommended to supplement the activity under simulated sunlight or pure visible light.

Reply: We thank the reviewer for pointing this out, enabling us to improve our manuscript. The relevant experiments proposed by the reviewers have been supplemented in the article. The ability to activate methane under visible light was detected by the AQY.

It has been modified in manuscript and SI:

Page 7: The apparent quantum yield (AQY) was further calculated to demonstrate the methane conversion capability of this catalyst (Table S8). The AQY of 873K-Nb₂O₅ for methane conversion is 0.43% under 365 nm light irradiation and significantly decreases under the visible light irradiation.

Table S8. Results of AQY for methane conversion at different wavelengths.

Wavelength (nm)	Produced hydrogen (μmol)	Light power density I (mW cm ⁻²)	AQY (%)
365	58.8	22	0.43
420	27.4	115	0.033
475	1.68	212	0.00042

Comment 2: Some basic experimental descriptions are missing, such as light intensity, reaction temperature, effective working wavelength range, etc. It is recommended to supplement relevant information in the revised manuscript.

Reply: We thank the reviewer for pointing this out, enabling us to improve our manuscript. The relevant experiments details proposed by the reviewers have been supplemented in the article.

It has been modified in SI:

Nonoxidative Coupling of Methane reaction test:

1. Batch quartz reactor reaction conditions: First, the catalysts were evacuated in a tube furnace at 393 K in a vacuum environment to remove the adsorbed water and other molecules. 5 mg catalysts were laminated to a closed quartz reactor (45 cm³, photoirradiation area, 28.27 cm²), then the reactor was evacuated for 10 min to remove air. 45 mL of pure methane (99.99%) was injected into the reactor by a gas injection needle and the reactor was placed in dark condition for 1 h to achieve an adsorption-desorption balance. The reactor was irradiated by a 300 W Xe lamp with 2000mW/cm² optical power density for 4 h. The methane conversion

proceeded under atmospheric pressure and without additional heating. The light band of the lamp: 200–2500 cm^{-1} . Reaction temperature, 67°C. The photoirradiation area, 28.27 cm^2 . The hydrocarbon products were extracted by the gas injection needle then analyzed by gas chromatography (GC) with a flame-ionization detector (FID). Hydrogen was analyzed by GC with a high-sensitivity thermal conductivity detector (TCD). For a long time reaction, the production was collected every 4 hours, replenish to atmospheric pressure with Ar gas after sampling. For cycle reaction, the sample repeated the vacuum activated after reaction to ensure that the adsorbed gas molecules were removed before proceeding to the next test.

2. Mobile-type reactor reaction conditions: The catalyst powder was pressed under 40 MPa pressure and ground into 40–60 mesh. Photocatalyst in a quartz cell, 0.05 g; photoirradiation area, 28.26 mm^2 ; cell volume, 113.04 mm^3 ; feed gas, 99.999% of CH_4 in flow rate 10 mL min^{-1} ; SV: 55000 h^{-1} ; light intensity, 2000 mW cm^{-2} , reaction temperature, 91°C.

Comment 3: The authors explored the effect of phase transition extent on the formation of FLP, and the activation effect of FLP on methane is emphasized in the mechanism discussion section. Is the FLP more important for methane activation or the defects created by the phase transition?

Reply: Thanks for your comment. First of all, the defects generated during the phase transition process are conducive to the improvement of photocatalytic activity. We have discussed this part in the Photocatalytic Performance part of the article. However, H_2 -treated samples can also generate defects such as oxygen vacancies, but the improvement in activity is much lower than that of NaBH_4 -treated samples. This suggests that FLP is obviously more important to the promotion of activity. Second, too many defects may also lead to the recombination of photogenerated electrons and holes, which makes the highest activity of defects only in a certain concentration range. Finally, the low-temperature phase transition process is conducive to the generation of adjacent oxygen vacancies and surface hydroxyl groups during the lattice twisting and stretching process, but an over high temperature cannot efficiently produce FLPs due to the severe elimination of hydroxyls. The appropriate degree of phase transition and construction of the highest concentration of FLP sites can achieve the highest methane conversion activity.

Comment 4: It should be pointed out that besides the coupled LA and LB mechanism proposed by the authors, the contribution from the Nb-OH group alone should not be negligible, which may generate active hydroxyl radicals to activate methane. This had been demonstrated to some extent by the quenching experiments using pyridine and pyrrole, respectively. It can be seen that after the neutralization of acidic sites, the catalyst still maintained nearly half of the original activity; however, when the basic sites are neutralized, the activity was greatly reduced.

Reply: We thank the reviewer for pointing this out enabling us to improve our manuscript. The methane TPD experiment after quenching at different sites was supplemented to study the adsorption capacity of methane. The NH_3 -TPD analysis confirmed the base site further promotes the acidity of the $\text{Nb}_{\text{L}}\text{V}$ site. Since quenching experiment cannot ensure that only a single acidic site or basic site remains, for the sake of rigor, we only discuss the mutual synergistic promotion between the two sites.

It has been modified in manuscript:

Page 7-8: To verify the effect of site quenching, CH_4 -temperature-programmed desorption (TPD) measurement evaluated the differences in methane adsorption,⁴⁴ and explained the affinity of acid/base site toward methane molecule and the trend of catalytic activities (Fig 2f and Table S12). The broad desorption bands can be observed in the CH_4 -TPD profiles of different catalysts. The wide gas desorption peaks at middle temperature range (200–400 °C)

correspond to chemical adsorption. Except for 873K-Nb₂O₅, most of the methane desorbs in a main peak around 250 °C, and there is a trace amount desorbing around 330 °C. While the chemical desorption peak area in higher temperature range increases significantly as compared with 873K-Nb₂O₅, indicating that more CH₄ is firmly adsorbed on the surface with FLP sites by chemisorption. The additional peak around 400 °C indicates double sites can further enhance the CH₄ adsorption capacity through increasing the binding strength for CH₄. The desorption yield of methane of 873K-Nb₂O₅ is evident s higher than single Vo site and 873K-Nb₂O₅ after acid/base quenched. This trend of the amount of adsorbed methane was well matched with the trend of the methane activation after site quench showed in Fig 2e, suggesting that methane adsorption behavior should be closely related to the methane activation.

Page 10: NH₃-TPD was further used to reveal the effect of calcination temperature on the acidic characteristics of NaBH₄-treated Nb₂O₅ (Fig 3d). The pristine T-Nb₂O₅ has negligible acid sites for NH₃ adsorption, while all samples treated with NaBH₄ show two distinct peaks above 600 °C attributed to strong acid sites. The H₂-Nb₂O₅ only shows a less intense peak at the lower temperature, suggesting a weaker acidity. The relation between the reduction temperature and acidity is confirmed by pyridine-FTIR (Fig 3e, Table S14). Through peak fitting of NH₃-TPD spectra (Table S15), it seems that sample 873K-Nb₂O₅ with the most distorted structure has the highest concentration of acid sites. It is also noted the acidity trend of NaBH₄-treated Nb₂O₅ is accordant with the change of Vo concentration. NH₃-TPD was also used to verify the quenching of acid sites through pyridine treatment. The results proved that the strongly acidic sites were indeed covered since the adsorption capacity of the sample for ammonia decreases significantly (Fig S20), which is accordant with the CH₄-TPD results.

Comment 5: Fig 3a exhibited the variation of Nb-O bond distance with the reduction temperature. Except for the change of the main peak, it can be noted that the 773K-Nb₂O₅ begin to show the broad peak around 2-3 Å. What does the peak change in this range mean?

Reply: There are two main reasons for this broad, difficult to identified peak: 1. Niobium oxide has a highly asymmetric crystal structure and a wide range of Nb-O bond lengths. This broad range of Nb-O bond lengths is averaged and reflected in the broader peak. 2. The severe disarrangements occur of the Nb local structure during the phase transformation. The rapid decline of oscillations frequency results in a broad peak in the FT-EXAFS spectra (Fig 3a).

Comment 6: The schematic lines drawn by the author in Fig 1b do not seem to be precisely aligned with the lattice fringes. Please check it carefully.

Reply: We thank the reviewer for pointing this out enabling us to improve our manuscript. The figure has been revised.

It has been modified in manuscript:

Comment 7: The methane conversion rate is used as the standard for activity comparison in the article. However, it can be seen from Figure 2a that the hydrogen yield from sample 873K-Nb₂O₅ becomes higher, so the selectivity to different alkane products should be supplemented.

Reply: We thank the reviewer for carefully reading our manuscript. In order to explore the composition of the product more accurately, we performed a combined gas chromatography-mass spectrometry detection of the product in the batch reactor for 4 hours. It can be seen that through mass spectrometry and detection with higher sensitivity, during the continuous reaction of the batch reactor, the main product will undergo further coupling to produce multi-carbon alkanes. This process leads to a large increase in the detected hydrogen. We supplemented the ethane product selectivity for the 4 h reaction and the selectivity for the ratio of ethane to hydrogen in a flow reactor. The reaction in the flow reactor benefits from the rapid discharge of the product, which presents ethane and hydrogen in an equimolar ratio.

It has been modified in manuscript:

Page 6-7: To further demonstrate the ability of catalysts to activate methane, the NOCM reaction in a flow reactor was carried out (Fig 2b, Fig S11 and Table S7). After a quick induction

period, the 873K-Nb₂O₅ has a stable ethane production rate of around 22.52 $\mu\text{mol h}^{-1}$ without significant decrease within 60 h. The ethane and hydrogen show a stable equimolar ratio during the reaction.

It has been modified in SI:

Table S3 Results of various samples for photocatalytic NOCM^[a]

Rate($\mu\text{mol/g h}$)	H ₂	Er \pm	C ₂ H ₆	Er \pm	C ₃ H ₈	Er \pm	C ₄ H ₁₀	Er \pm	CH ₄ conversion	Er \pm	Ethane selectivity
Nb ₂ O ₅	29.7	4.1	32.3	3.5	0.7	0.5	0.0	0.0	66.8	5.8	97.9
H ₂ -Nb ₂ O ₅	249.1	7.4	217.2	14.2	13.0	1.0	2.6	0.6	484.1	32.8	93.3
673K-Nb ₂ O ₅	319.3	8.6	284.2	9.4	12.4	1.5	2.2	0.3	614.5	21.9	95.1
773K-Nb ₂ O ₅	351.3	13.5	353.9	13.3	15.8	1.7	3.2	1.5	767.8	24.4	94.9
873K-Nb ₂ O ₅	780.9	8.6	600.8	4.3	62.3	1.1	17.0	1.1	1456.5	11.2	88.3
973K-Nb ₂ O ₅	567.4	5.3	535.5	6.5	25.6	1.3	5.6	1.1	1170.1	5.4	94.5

[a] Reaction conditions were as follows: samples, 0.005 g; reaction time, 4 h; reactant, 45 mL of methane; light source, 300 W Xe lamp; reaction temperature, room temperature; quartz reactor, 45 cm³. The products were analyzed by gas chromatography with flame-ionization and thermal conductivity detector.

Comment 8: Fig 2a shows the UV-vis DRS spectra of different Nb₂O₅ samples. While the absorption of visible light is enhanced, the absorption of ultraviolet light is weakened on the NaBH₄-treated samples. For this sample, if the AQY results from the ultraviolet and visible light irradiation can be provided, the contribution of different wavelengths can be better understood.

Reply: We thank the reviewer for pointing this out enabling us to improve our manuscript. Related experiments have been supplemented in the manuscript.

It has been modified in manuscript:

Page 7: To further demonstrate the ability of catalysts to activate methane, the NOCM reaction in a flow reactor was carried out (Fig 2b, Fig S11 and Table S7). After a quick induction period, the 873K-Nb₂O₅ has a stable ethane production rate of around 22.52 $\mu\text{mol h}^{-1}$ without significant decrease within 60 h. The ethane and hydrogen show a stable equimolar ratio during the reaction. The apparent quantum yield (AQY) was further calculated to demonstrate the methane conversion capability of this catalyst (Table S8). The AQY of 873K-Nb₂O₅ for methane conversion is 0.43% under 365 nm light irradiation and significantly decreases under the visible light irradiation.

To understand the enhanced mechanism over NaBH₄-treated Nb₂O₅, the spectroscopic and photoelectric characteristics of different samples were further analyzed. Compared with the pristine and H₂-treated Nb₂O₅ only with absorption in the UV region, NaBH₄-treated Nb₂O₅ shows the photoresponse from UV to near-infrared light regions (Fig 2c). The photoluminescence (PL) emission, time-resolved fluorescence decay, and transient photocurrent response analyses (Fig S13-15 and Table S10) show that the 873K-Nb₂O₅ has a good photo-responsive performance. However, there is no obvious difference among different NaBH₄-treated Nb₂O₅ samples with similar absorption bands, which suggests that the spectroscopic and photoelectric properties are not the main reason for the enhanced NOCM. This result is accordant with the significantly decreased activity of 873K-Nb₂O₅ in visible light region, confirming the enhanced photocatalytic NOCM activity is not only caused by the

expanded light absorption.

Table S8. Results of AQY for methane conversion at different wavelengths.

Wavelength (nm)	Produced hydrogen (μmol)	Light power density I (mW cm^{-2})	AQY (%)
365	58.8	22	0.43
420	27.4	115	0.033
475	1.68	212	0.00042

Response to Reviewer 3

General comments: The authors studied the non-oxidative coupling of methane (NOCM) over several Nb₂O₅ based materials and found that 873K-Nb₂O₅ sample has high photocatalytic activity of 1,456 μmol/g.h. Based on several spectroscopic observation and theoretical calculations, they claimed that frustrated Lewis pairs consisting of low-valence Lewis acid (LA) and Lewis base (LB) play key roles in the enhancement of photocatalytic NOCM. While interesting, there are three problems in the manuscript at the present stage.

Reply: We would like to thank the reviewer for the careful reading of our manuscript. The suggestions and comments have been carefully considered and we made the corresponding revision to address the points the reviewer has raised, as shown below. We hope our revision can make the paper much more acceptable for publication in *Nature Communications*.

Comment 1: The experimental data presented here do not show any evidence that methane can interact strongly with these surfaces with frustrated Lewis pairs and vacancies. Although IR measurement of Py and TPD of NH₃ shows the existence of frustrated Lewis pairs on their samples, there is no direct experimental evidence that the methane, one of the extremely weakly adsorbed molecular species, does indeed interact with these surface sites and show polarization. If methane molecules strongly interact with their sample surface as proposed in their calculation, then, adsorbed methane is clearly detected by the infrared absorption spectroscopy. It was reported that C-H stretching mode of methane derived from the weakened C-H bonds typically redshifts from the corresponding gas-phase values by ~200 cm⁻¹ [Table 1 of Catal Today 160, 213 (2011)]. Therefore, the existence of the softened C-H vibrational peak of the strongly adsorbed methane species should be demonstrated by IR spectroscopy for these samples under both dark and light irradiation conditions, and the correlation with reaction activity and consistency with the mechanism proposed by their theoretical calculation should be verified carefully. Based on these additional results and arguments, authors should validate their claims and mechanism.

Reply: We thank the reviewer for pointing this out, enabling us to improve our manuscript. We consulted related literature on softened C-H bond vibrations, and found that the redshifts of vibrational modes on the surface of different metals or metal oxides range from tens to more than 200 cm⁻¹. We conducted related experiments according to the reviewers' comments and added the results to the article. For the pristine Nb₂O₅, the ability to adsorb and activate methane is weak in the dark. The 873K-Nb₂O₅ with abundant FLP sites has the ability to adsorb methane under the dark conditions, and the adsorption to methane is significantly enhanced under light irradiation. The weakened vibration peak of methane C-H appeared after light irradiation, which confirmed our previous mechanism. We have cited the reference Catal Today 160, 213 (2011)].

It has been modified in manuscript:

Page 15:

In situ infrared absorption spectroscopy was used to verify that methane can strongly interact with FLP. There have been many studies exploring the softened C-H vibrational on the surface of different metal oxides surface.^{50,51,52,53,54} Fig S26 shows in situ FTIR spectra after exposing pristine Nb₂O₅ and NaBH₄-treated 873K-Nb₂O₅ to pure methane under dark and light environment. The weakened C-H bond stretching mode typically redshifts from the

corresponding gas-phase values (Table S24), and the 873K-Nb₂O₅ with more polarized environment under light excitation exhibited the band at 2822 cm⁻¹ appear from the softened stretching vibration ν_1 , which is an infrared forbidden mode in the free CH₄ molecules.⁵⁵ This peak attributed to soft vibrations confirm more polarized methane activation under photoexcitation. The activation of the ν_1 mode and the frequency redshift provides the reduction of symmetry of methane over 873K-Nb₂O₅.⁵⁰ There is no difference before and after adsorption of methane and light exposure over pristine Nb₂O₅, which confirms that FLP participates in the polarization of methane and induce the C-H bond vibrational mode change.

Comment 2: Related to the above issue, differences in the interaction of each sample with methane should be directly evaluated as differences in methane adsorption energy by adsorption isotherm and/or TPD measurements [J.Chem.Phys.132,024709(2010)]. It is highly desirable that these basic data is discussed in relation to the photocatalytic effects of the frustrated Lewis pairs and vacancies for each sample.

Reply: We thank the reviewer for pointing this out, enabling us to improve our manuscript. We have supplemented relevant experiments and performed the analysis as suggested by the reviewers. Through the CH₄-TPD experiment, it can be verified that the double-site synergy can promote the adsorption activation of methane, and the adsorption capacity of methane is stronger. The Reference[J.Chem.Phys.132,024709(2010)] has been cited in our revised manuscript.

It has been modified in manuscript:

Page 8: To verify the effect of site quenching, CH₄-temperature-programmed desorption (TPD) measurement evaluated the differences in methane adsorption,⁴⁴ and explained the affinity of acid/base site toward methane molecule and the trend of catalytic activities(Fig 2f and Table S12). The broad desorption bands can be observed in the CH₄-TPD profiles of different catalysts. The wide gas desorption peaks at middle temperature range (200-400 °C) correspond to chemical adsorption. Except for 873K-Nb₂O₅, most of the methane desorbs in a main peak around 250 °C, and there is a trace amount desorbing around 330 °C. While the chemical desorption peak area in higher temperature range increases significantly as compared with 873K-Nb₂O₅, indicating that more CH₄ is firmly adsorbed on the surface with FLP sites by chemisorption. The additional peak around 400 °C indicates double sites can further enhance the CH₄ adsorption capacity through increasing the binding strength for CH₄. The desorption yield of methane of 873K-Nb₂O₅ is evident s higher than single Vo site and 873K-Nb₂O₅ after acid/base quenched. This trend of the amount of adsorbed methane was well matched with the trend of the methane activation after site quench showed in Fig 2e, suggesting that methane adsorption behavior should be closely related to the methane activation.

Comment 3: Although authors would want to emphasize methane conversion under mild conditions as in line 88, they did not indicate and discuss the effect of temperature rise and methane pressure under the irradiation of 300 W Xe lamp. In general, sample temperatures substantially rise under the irradiation of such intense light. Can the authors really claim that the reaction is occurring under mild conditions?

Reply: We understand reviewers' concerns. All our reactions were performed under

atmospheric pressure and without additional heating. The batch quartz reactor has good ability for heat dissipation, and the temperature at the bottom of the reactor was detected to be 67 °C. According to another reviewer's comments, we supplemented the experiment in the flow reactor, which is made of stainless steel. The temperature detection rod detects that the temperature at the bottom of the catalyst gasket inside the flow-type reactor is 91 °C.

It has been modified in manuscript:

This work provides guidance for the rational design and construction of photocatalyst in a highly polarized environment for efficient methane conversion under atmospheric pressure and without additional heating.

It has been modified in SI:

Nonoxidative Coupling of Methane reaction test:

1. Batch quartz reactor reaction conditions: First, the catalysts were evacuated in a tube furnace at 393 K in a vacuum environment to remove the adsorbed water and other molecules. 5 mg catalysts were laminated to a closed quartz reactor (45 cm³, photoirradiation area, 28.27 cm²), then the reactor was evacuated for 10 min to remove air. 45 mL of pure methane (99.99%) was injected into the reactor by a gas injection needle and the reactor was placed in dark condition for 1 h to achieve an adsorption-desorption balance. The reactor was irradiated by a 300 W Xe lamp with 2000mW/cm² optical power density for 4 h. The methane conversion proceeded under atmospheric pressure and without additional heating. The light band of the lamp:200~2500 cm⁻¹. Reaction temperature,67°C. The photoirradiation area, 28.27 cm²The hydrocarbon products were extracted by the gas injection needle then analyzed by gas chromatography (GC) with a flame-ionization detector (FID). Hydrogen was analyzed by GC with a high-sensitivity thermal conductivity detector (TCD). For a long time reaction, the production was collected every 4 hours, replenish to atmospheric pressure with Ar gas after sampling. For cycle reaction, the sample repeated the vacuum activated after reaction to ensure that the adsorbed gas molecules were removed before proceeding to the next test.

2. Mobile-type reactor reaction conditions: The catalyst powder was pressed under 40 MPa pressure and ground into 40–60 mesh. Photocatalyst in a quartz cell, 0.05 g; photoirradiation area, 28.26 mm²; cell volume, 113.04 mm³; feed gas, 99.999% of CH₄ in flow rate 10 mL min⁻¹; SV:55000 h⁻¹; light intensity, 2000 mW cm⁻², reaction temperature, 91°C.

Comment 4: TEM image (Fig. 1b) is blurred and unclear. Most of the figure texts are also blurred and not clear.

Reply: We thank the reviewer for pointing this out. The corresponding figures have been corrected. Some pictures may lose clarity due to compression during uploading, and we will upload high-definition pictures and images later.

Fig 1b

Reference

44. Weaver JF, Hakanoglu C, Hawkins JM, Asthagiri A. Molecular adsorption of small alkanes on a PdO (101) thin film: Evidence of σ -complex formation. *The Journal of chemical physics* **132**, 024709 (2010).
50. Ferrari AM, Huber S, Knözinger H, Neyman KM, Rösch N. FTIR Spectroscopic and Density Functional Model Cluster Studies of Methane Adsorption on MgO. *The Journal of Physical Chemistry B* **102**, 4548-4555 (1998).
51. Li C, Li G, Xin Q. FT-IR spectroscopic studies of methane adsorption on magnesium oxide. *The Journal of Physical Chemistry* **98**, 1933-1938 (1994).
52. Weaver JF, Hinojosa JA, Hakanoglu C, Antony A, Hawkins JM, Asthagiri A. Precursor-mediated dissociation of n-butane on a PdO(101) thin film. *Catalysis Today* **160**, 213-227 (2011).
53. Koitaya T, Ishikawa A, Yoshimoto S, Yoshinobu J. C-H Bond Activation of Methane through Electronic Interaction with Pd (110). *The Journal of Physical Chemistry C* **125**, 1368-1377 (2021).
54. Weaver JF, Hinojosa Jr JA, Hakanoglu C, Antony A, Hawkins JM, Asthagiri A. Precursor-mediated dissociation of n-butane on a PdO (1 0 1) thin film. *Catalysis today* **160**, 213-227 (2011).
55. MILLEN D. HERZBERG, G-MOLECULAR SPECTRA AND MOLECULAR STRUCTURE.). MACMILLAN MAGAZINES LTD PORTERS SOUTH, 4 CRINAN ST, LONDON N1 9XW, ENGLAND (1967).

REVIEWERS' COMMENTS

Reviewer #1 (Remarks to the Author):

The authors well revised the manuscript according to the comments from the reviewer with additional experimental results. The newly obtained results are very valuable. This manuscript is now recommendable for the publication in this journal after adequate minor revisions as listed below:

1. The wrong manners for preparing graphs still appeared:

- Figure 3c, the lower covers should not start at minus two.
- Fig. S13, show the values for the vertical axis, especially show zero.
- Fig. S16a and b, show the values for the vertical axis, especially show zero and one (normalized to be unity).
- Fig. S18, the lower covers should not start at minus two.
- Fig. S20, show the values for the vertical axis, especially show zero. Here, we want to compare right and left figures quantitatively. Show something that indicate the values.
- Fig. S26, the baseline for every curve would not start these values indicated at the vertical axis. Show them in the same manner as Figure 1a.

2. Show the units correctly for the values listed in Table S3 (%),

Reviewer #2 (Remarks to the Author):

I thank the authors for their edits and response. They have addressed my concerns. Similar to the first reviewer's comments, please further supplement the important references for photocatalytic NOCM reaction (Nat. Energy 2022, 7, 1042; Nat. Commun. 2022, 13, 2806; Angew. Chem. Int. Ed . 2021, 60, 20760; Angew. Chem. Int. Ed. 2011, 50, 8299).

Reviewer #3 (Remarks to the Author):

The reviewer's sincere effort to revise the manuscript is greatly appreciated. In response to the original comment 1, the authors present additional vibrational spectroscopy data in Fig. S26. Although this data does not demonstrate the existence of FLP sites that strong adsorption and polarization of methane occurs predominantly on the 873K-Nb2O5 sample under light non-irradiation conditions, the authors also showed the CH4-TPD data that indicates strong adsorption of methane for this sample. After the adsorption energy of methane on these samples is carefully evaluated from this TPD data, this work would be acceptable as paper in Nature Communications.

<Minor Comment>

Not only non-oxidative coupling but also reactions in the presence of water vapor are interesting systems for methane conversion reactions[Communications Chemistry 6, 8 (2023)]. The referee would like to suggest you to increase the impact of this paper by mentioning in the final part of the manuscript how the authors think about the possibility of the application of the FLP site in other methane conversion photocatalyst system in which other molecules such as adsorbed water are involved in the methane conversion reactions.

Reply to the comments of Reviewers

Response to Reviewer 1

General comments: The authors well revised the manuscript according to the comments from the reviewer with additional experimental results. The newly obtained results are very valuable. This manuscript is now recommendable for publication in this journal after adequate minor revisions as listed below:

Reply: We would like to thank the reviewer for the careful reading of our manuscript. According to your suggestion, we have revised the figures and tables mentioned in the review. The corresponding revisions and supplementary data are highlighted in the revised manuscript and shown below. We hope our revision can make the manuscript more acceptable for publication in Nature Communications.

Comment 1: The wrong manners for preparing graphs still appeared:

- Figure 3c, the lower covers should not start at minus two.
- Fig. S13, show the values for the vertical axis, especially show zero.
- Fig. S16a and b, show the values for the vertical axis, especially show zero and one (normalized to be unity).
- Fig. S18, the lower covers should not start at minus two.
- Fig. S20, show the values for the vertical axis, especially show zero. Here, we want to compare right and left figures quantitatively. Show something that indicate the values.
- Fig. S26, the baseline for every curve would not start these values indicated at the vertical axis. Show them in the same manner as Figure 1a.

It has been modified in manuscript and SI:

Figure 3. Analysis of FLP sites. (a) FT $k^3\chi(R)$ Nb K-edge EXAFS of different Nb₂O₅ samples. (b) Fourier transformed EXAFS fitting results of the Nb-O coordination numbers (CNs) for different samples. Error bars represent standard deviation. (c) Nb K-edge EXAFS (points) and curvefit (line) for 873K-Nb₂O₅, shown in k^3 -weighted R -space (FT magnitude and imaginary components). The data are k^3 -weighted and not phase-corrected. (d) NH₃-TPD spectra of samples. (e) Pyridine-IR spectra of different NaBH₄-treated Nb₂O₅. (f) Relationship between the surface hydroxyl group and oxygen vacancy concentration of different samples.

Supplementary Figure 13. Room-temperature PL emission spectra (excitation wavelength is 350 nm) of different Nb_2O_5 samples.

Supplementary Figure 16. Normalized Nb K-edge XANES spectra of different Nb_2O_5 samples. The position of the dotted line in b is the highest point of corresponding first derivative absorption curves of Nb_2O_5 .

Supplementary Figure 18. Nb K-edge EXAFS (points) and curvefit (line) for (a) T- Nb_2O_5 , (b) H_2 - Nb_2O_5 and (c-e) NaBH_4 -treated Nb_2O_5 in different temperature shown in k^3 -weighted R -space (FT magnitude and imaginary component). The data are k^3 -weighted and not phase-corrected.

Supplementary Figure 20. NH₃-TPD spectra of samples before and after pyridine quenching.

Supplementary Figure 26. The in situ diffuse reflectance infrared Fourier transform spectroscopy of (a) pristine Nb₂O₅ and (b) 873K-Nb₂O₅ under dark and light situation. (20% CH₄ gas 20 sccm, 298 K in dry condition).

Comment 2: Show the units correctly for the values listed in Table S3 (%),

It has been modified in SI:

Supplementary Table 3 Results of various samples for photocatalytic NOCM^[a]

Rate(μmol/g h)	H ₂	Er±	C ₂ H ₆	Er±	C ₃ H ₈	Er±	C ₄ H ₁₀	Er±	CH ₄ conversion	Er±	Ethane selectivity
Nb ₂ O ₅	29.7	4.1	32.3	3.5	0.7	0.5	0.0	0.0	66.8	5.8	97.9%
H ₂ -Nb ₂ O ₅	249.1	7.4	217.2	14.2	13.0	1.0	2.6	0.6	484.1	32.8	93.3%
673K-Nb ₂ O ₅	319.3	8.6	284.2	9.4	12.4	1.5	2.2	0.3	614.5	21.9	95.1%
773K-Nb ₂ O ₅	351.3	13.5	353.9	13.3	15.8	1.7	3.2	1.5	767.8	24.4	94.9%
873K-Nb ₂ O ₅	780.9	8.6	600.8	4.3	62.3	1.1	17.0	1.1	1456.5	11.2	88.3%
973K-Nb ₂ O ₅	567.4	5.3	535.5	6.5	25.6	1.3	5.6	1.1	1170.1	5.4	94.5%

[a] Reaction conditions were as follows: samples, 0.005 g; reaction time, 4 h; reactant, 45 mL of methane; light source, 300 W Xe lamp; reaction temperature, room temperature; quartz reactor, 45 cm³. The products were analyzed by gas chromatography with flame-ionization and thermal conductivity detector.

Response to Reviewer 2

General comments: I thank the authors for their edits and response. They have addressed my concerns. Similar to the first reviewer's comments, please further supplement the important references for photocatalytic NOCM reaction (Nat. Energy 2022, 7, 1042; Nat. Commun. 2022, 13, 2806; Angew. Chem. Int. Ed. 2021, 60, 20760; Angew. Chem. Int. Ed. 2011, 50, 8299).

Reply: We appreciate the reviewer for their generous recommendation of our manuscript for publication in *Nature Communications*. There are many interesting pioneer studies for photocatalytic NOCM reaction, and we have supplemented some related references to give readers a more comprehensive background introduction.

It has been modified in manuscript:

7. Li L, *et al.* Efficient sunlight-driven dehydrogenative coupling of methane to ethane over a Zn⁺-modified zeolite. *Angewandte Chemie International Edition* **50**, 8299-8303 (2011).

8. Zhang L, *et al.* Visible-light-driven non-oxidative dehydrogenation of alkanes at ambient conditions. *Nature Energy* **7**, 1042-1051 (2022).

9. Zhang W, *et al.* High-performance photocatalytic nonoxidative conversion of methane to ethane and hydrogen by heteroatoms-engineered TiO₂. *Nature Communications* **13**, 2806 (2022).

10. Wang G, *et al.* Light-induced nonoxidative coupling of methane using sSupplementary Table olid solutions. *Angewandte Chemie International Edition* **60**, 20760-20764 (2021).

Response to Reviewer 3

General comments: The reviewer's sincere effort to revise the manuscript is greatly appreciated. In response to the original comment 1, the authors present additional vibrational spectroscopy data in Fig. S26. Although this data does not demonstrate the existence of FLP sites that strong adsorption and polarization of methane occurs predominantly on the 873K-Nb₂O₅ sample under light non-irradiation conditions, the authors also showed the CH₄-TPD data that indicates strong adsorption of methane for this sample. After the adsorption energy of methane on these samples is carefully evaluated from this TPD data, this work would be acceptable as paper in Nature Communications.

<Minor Comment>: Not only non-oxidative coupling but also reactions in the presence of water vapor are interesting systems for methane conversion reactions[Communications Chemistry 6, 8 (2023)]. The referee would like to suggest you to increase the impact of this paper by mentioning in the final part of the manuscript how the authors think about the possibility of the application of the FLP site in other methane conversion photocatalyst system in which other molecules such as adsorbed water are involved in the methane conversion reactions.

Reply: We would like to thanks to the reviewers for their affirmation of our work and efforts. There are many interesting studies for photocatalytic methane conversion systems, and we have supplemented some related references to give readers a more comprehensive background introduction.

It has been modified in manuscript:

Non-oxidative coupling as a model reaction is beneficial to explicitly understanding the relationship between methane activation and light irradiation. Constructing a polarized environment should also be conducive to the improvement of the activity in other methane conversion reactions, such as methane dry reforming and methane steam reforming. This research demonstrates the principle of constructing a polarization environment for photocatalytic C-H activation of methane, providing a new perspective on the structural design of efficient photocatalysts for methane conversion without the assistance of precious metals.

15. Sato H, Ishikawa A, Saito H, Higashi T, Takeyasu K, Sugimoto T. Critical impacts of interfacial water on C-H activation in photocatalytic methane conversion. *Communications Chemistry* 6, 8 (2023).